



# Glaciers and ice caps through the Holocene: A pan–Arctic synthesis of lake–based reconstructions

Laura J. Larocca[1], Yarrow Axford[1]

[1]Department of Earth and Planetary Sciences, Northwestern University, 2145 Sheridan Road, Evanston, IL 60208 USA

*Correspondence to*: Laura J. Larocca (Laura@earth.northwestern.edu)

**Abstract.** The recent retreat of nearly all glaciers and ice caps (GICs) located in Arctic regions is one of the most clear and visible signs of ongoing climate change. This paper synthesizes published records of Holocene GIC fluctuations from lake archives, placing their recent retreat into a longer–term context. Our compilation includes sixty–six lake–based GIC records (plus one non–lake–based record from the Russian Arctic) from seven Arctic regions: Alaska; the archipelagos of the eastern Canadian Arctic; GICs peripheral to the Greenland Ice Sheet; Iceland; the Scandinavian peninsula; Svalbard; and the Russian high Arctic. For each region, and for the full Arctic, we summarize evidence for when GICs were smaller than today or absent altogether, indicating warmer than present summers, and evidence for when GICs regrew in lake catchments, indicating summer cooling. Consistent with orbitally driven high boreal summer insolation in the early Holocene, the pan–Arctic compilation suggests that the majority (50% or more) of studied GICs were smaller than present or absent by ~10 ka. The regional compilations suggest even earlier GIC loss, and thus warmth, in the Russian Arctic and in Svalbard. We find the highest percentage (>90%) of Arctic GICs smaller than present or absent in the middle Holocene ~7–6 ka, probably reflecting more spatially ubiquitous and consistent summer warmth during this period than in the early Holocene. Following this interval of widespread warmth, our compilation shows that GICs across the Arctic began to regrow, and summers began to cool by ~6 ka. Together, the pan–Arctic records also suggest two periods of enhanced GIC growth in the mid–to–late Holocene, from ~4.5–3 ka and after ~2 ka. The regional records show substantial variability in the timing of GIC regrowth within and between regions, suggesting that the Arctic did not cool synchronously despite the smooth and hemispherically symmetric decline in Northern Hemisphere summer insolation. In agreement with other studies, this implies a combined response to glacier–specific characteristics such as topography, and to other climatic forcings and feedback mechanisms, perhaps driving periods of increased regional cooling. Today, the direction of orbital forcing continues to favor GIC expansion, however, the rapid retreat of nearly all Arctic GICs underscores the current dominance of anthropogenic forcing on GIC mass balance. Our review finds that in the first half of the Holocene, most of the Arctic's small GICs became significantly reduced or melted away completely in response to summer temperatures that, on average, were only moderately warmer than today. In comparison, future projections of temperature change in the Arctic far exceed estimated early Holocene values in most locations, portending the eventual loss of most of the Arctic's small GICs.

## 1 Introduction

Globally, mass loss from glaciers and ice caps (GICs) is accelerating (Hugonnet et al., 2021). Between 2000–2019, GICs worldwide lost a mass of 267±16 gigatonnes per year, which is equivalent to 21±3% of the observed sea–level



rise (Hugonnet et al., 2021). Notably, the roughly 50,000 GICs located in Arctic regions accounted for ~70% of this recent loss (Hugonnet et al., 2021, extended data Table 1; Vaughan et al., 2013). By the end of the century, mean surface air temperature in the Arctic is expected to warm by 2.2–8.3°C—a rate that is amplified relative to the global mean (Collins et al., 2013, temperature anomalies relative to the 1986–2005 reference period). Accordingly, regionally

differentiated, global–scale projections of GIC mass change find Arctic GICs to be the largest contributors to forecasted global ice volume loss by 2100 (Radić et al., 2011; 2013). The continued wastage of Arctic GICs is expected to have a myriad of sociocultural and economic ramifications for Arctic communities, including major alterations to hydrological systems at a local scale, potentially affecting water availability, quality, and downstream aquatic ecosystems (Huntington et al., 2019; Pachauri et al., 2008).


The current and projected rapid changes to the Arctic cryosphere are even more striking when considered within a longer–term context (e.g., Kaufman et al., 2009; Fisher et al., 2012; Miller et al., 2013). GIC fluctuations over the Holocene have been reconstructed using a mix of discontinuous and continuous proxies including the mapping and dating of glacial moraines, lichenometry and tree ring records, and from proglacial lake and speleothem records

(Solomina et al., 2015). Overall, these records indicate that many Arctic GICs were small or had completely melted away in the early–to–middle Holocene in response to orbitally forced summer warmth in the Northern Hemisphere (Solomina et a., 2015). These records also show that many of the GICs melting away today reformed during the middle–to–late Holocene as summer temperature cooled from the insolation–driven Holocene maximum (McKay et al., 2018; Solomina et al., 2015).  Recent, anthropogenic–driven warming has sharply reversed this long–term,

insolation–driven, cooling trend and GIC expansion, and is expected to cut short the lifespans of Arctic GICs that in many cases have existed for several thousand years.

While relatively widely studied, many questions linger about Holocene climate. Globally, climate simulations and proxy reconstructions fundamentally disagree on the overall direction of mean annual temperature trends through the

Holocene, a significant data–model discrepancy coined the *Holocene temperature conundrum* (Liu et al., 2014). Furthermore, to date, there have been few published multi–proxy syntheses of Holocene climate specific to the Arctic and most have been focused on subregions or narrower time periods (e.g., Kaufman et al., 2004, 2009; Kaplan and Wolfe, 2006; Briner et al., 2016; McKay et al., 2018; Axford et al., 2020). Although chiefly driven by symmetrical orbital forcing, synthesis studies of temperature sensitive proxy data suggest that the timing and magnitude of the

Holocene Thermal Maximum (HTM) was spatially and temporally asymmetrical across the Arctic (e.g., Kaufman et al., 2004; Kaplan and Wolfe, 2006; Briner et al., 2016). Likewise, the onset and rate of summer cooling in the Arctic in the middle–to–late Holocene did not occur homogenously (McKay et al., 2018). These syntheses reaffirm the notion that the Arctic does not behave as a distinct climatological unit and that the climatic responses to insolation and other forcings through the Holocene were complex (McKay et al., 2018). An improved understanding of the regional and

Arctic–wide patterns of multi–millennial Holocene temperature changes may elucidate the driving factors that control regional climates, and thus, help anticipate the local–scale consequences of future Arctic warming.



Recent observations have confirmed that GICs respond sensitively and quickly, on decadal timescales, primarily to changes in summer temperature and to a lesser extent, accumulation season precipitation (Oerlemans, 2005; Koerner, 2005; Bjørk et al., 2012). Sediment records from glacial lakes offer invaluable continuous archives of GIC variations over the Holocene, recording their presence and absence on the landscape, and in some cases more subtle variations in GIC size over time. Records of GIC fluctuations from lake sediments can be considered a relatively straightforward qualitative proxy for summer temperature. GICs' exceptional sensitivity to modest summer temperature changes make their individual archives important indicators of regional climate, and their combined archives an important gauge of broad, large–scale climate trends through the Holocene (Kelly and Lowell, 2009; Solomina et al., 2015). Here, we synthesize published lake–based GIC records (n=66, plus one non–lake–based record from Franz Josef Land, included due to a dearth of records from the Russian Arctic) to assess regional– and pan–Arctic–scale summer temperature trends through the Holocene. Our review covers seven geographical regions above 58°N from which such records are available: Alaska; the archipelagos of the eastern Canadian Arctic; GICs peripheral to the Greenland Ice Sheet; Iceland; the Scandinavian peninsula; Svalbard; and the Russian high Arctic. For each region, we summarize evidence for when GICs were smaller than today, or absent altogether, documenting the timing of warmer than present summer conditions. In addition, we summarize evidence of GIC regrowth in lake catchments, documenting summer cooling, and specifically, when equilibrium–line altitudes (ELAs) first lowered to intersect the local topography.

Our review focuses on the following questions: **1.** What are the regional and pan–Arctic trends in Holocene GIC fluctuations gleaned from lake–based records? **2.** When do lake–based GIC records indicate (a minimum bound on) the onset of warmer than present summers in each region and Arctic–wide? **3.** When do lake–based GIC records indicate the subsequent onset of GIC regrowth, and by inference summer cooling, in each region and Arctic–wide?



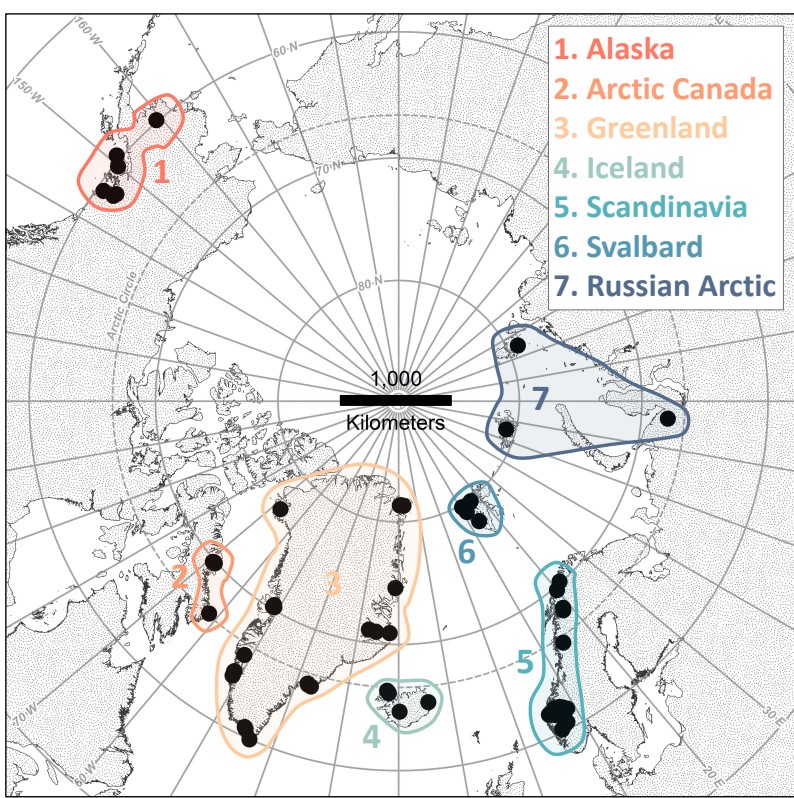

**Figure 1: Map of Holocene lake–based GIC records used in this study.** From west to east, 1. Alaska (n=6); 2. Arctic Canada (n=5); 3. Greenland (n=22); 4. Iceland (n=5); 5. Scandinavia (n=20); 6. Svalbard (n=6); 7. Russian Arctic (n=2, plus 1 non–lake record). For a list of included individual studies, see **Table S1** in Supplementary materials.


## 2 Data and approach

Arctic regions host roughly 50,000 GICs that account for nearly 60% of the global total glacierized area (Vaughan et al., 2013, Table 4.2). Very few extant GICs have multi–decadal in–situ mass–balance measurements (Zemp et al.,

2009). Similarly, prior to historical time, knowledge of Holocene GIC fluctuations is sparse, partly because recent advances have erased much of the geomorphic evidence of their earlier histories. However, glacial lake records offer valuable insight into past GIC variations and their sensitivity to climate changes. We compiled all published Holocene lake–based GIC records from the Arctic, which we define as land area above 58°N. The records report local GIC fluctuations through the Holocene reconstructed via analysis of lacustrine sediments from downstream glacial lakes.

Erosion at the ice–bed interface produces rock flour which is transported to downstream lakes via proglacial meltwater streams (Dahl et al., 2003). In general, intervals dominated by minerogenic, fine–grained (sandy and/or silty–clay) sediment are interpreted as reflective of glacier presence in a lake catchment, whereas intervals in which the sediment is higher in organic material (gyttja) are interpreted as periods when the glacier was reduced in size relative to today



or of no glacier in the catchment. In some studies, sediments from nearby non–glacial control lakes are also analyzed
to clarify the glacial signal (Dahl et al., 2003). We interpret periods in which GICs are reported as smaller than present
or absent altogether as indicative of summer temperatures warmer than present. In addition, we suggest that the lake–
based GIC evidence presented here represents a minimum bound on the onset of warmer than present summers for
several reasons. First, many of the lake records do not extend through the entire Holocene, are limited by the record's
length, and may not contain the onset of warmth. For instance, GIC absence from the start of a lake record can only
place a minimum bound on the onset of warmer than present conditions. Second, although relatively quick responders,
it takes some time for GICs to adjust and reach equilibrium or to melt away completely following a shift in climate.

In total we compiled 66 lake–based records of Holocene GIC variations from seven regions: Alaska (n=6), the
archipelagos of the eastern Canadian Arctic (n=5), GICs peripheral to the Greenland Ice Sheet (n=22), Iceland (n=5),
the Scandinavian peninsula (n=20), Svalbard (n=7), and the Russian high Arctic (n=2) (**Fig. 1**). We excluded
ambiguous records (that do not clearly define when GICs were smaller than present or absent, or when they regrew)
and included one non–lake–based study from the Russian Arctic due to the dearth of published glacial lake records
there. We used regional divisions defined by the Randolph Glacier Inventory (RGI6.0; RGI Consortium, 2017), but
considered the Ural Mountains area as part of the Russian Arctic (the Ural Mountain area is defined as part of North
Asia in RGI6.0). We also note that in Canada, all the available lake records are located within the region defined as
Arctic Canada South in the RGI. For each lake record we documented when GICs were smaller than present or had
melted away completely (indicating that summer temperatures were warmer than present) in the early–to–middle
Holocene and when the lake subsequently became glacially influenced again (indicating GIC regrowth and summer
cooling) in the middle–to–late Holocene.


The interpretation of individual glacial lake records is dependent upon the configuration of glaciers within the
catchments. In our review, we identified three common glacier–lake systems (**Fig. 2**). In some studies chains of
proglacial lakes are used, instead of a singular downstream lake. The most common system (glacier–lake system 1;
**Fig. 2**, left panel) allows for continuous reconstruction of GIC fluctuations over time, including GIC presence,
absence, and potentially more subtle variations in size. Sediment records from glacier–lake systems 2 and 3 are
considered threshold, "on–off" type records where the glacier or ice cap must breach a topographic boundary to input
glacial sediments into the lake. In glacier–lake system 2 (**Fig. 2**, middle panel), the present–day glacier is behind the
topographic boundary and glacial sediments will only be deposited when the glacier was larger than present. Thus,
organic–rich sediment deposition can be indicative of a glacier at present–day size, smaller than present, or absent
altogether. Additional evidence beyond the lake record is needed to distinguish between these three glacier states.
Finally, in glacier–lake system 3 (**Fig. 2**, right panel), the present–day glacier is beyond a topographic boundary and
presently inputs glacial sediments to the lake. In this case, organic–rich sediment deposition can be indicative of a
glacier smaller than present or absent. Again, additional evidence is needed to distinguish between these two glacier
states.






All ages that delimit GIC fluctuations are stated from the original publications, including original calibrations and any marine reservoir corrections, and are reported as thousands of years before 1950 AD (i.e., ka). Uncalibrated $^{14}$C ages in original publications were calibrated and reported here as the median probability using CALIB version 8.2 and the IntCal20 calibration curve (Reimer et al., 2020; Stuiver et al., 2021) for terrestrial samples and the Marine20

calibration curve (Heaton et al., 2020) for marine samples. To represent the GIC evidence we took a binary approach and defined GIC status in 100–year intervals from 12–0 ka, where 0 = glacially influenced, and 1 = a smaller than present or absent glacier or ice cap (see supplementary data). A small subset of the original published studies reconstruct more nuanced information on glacier variations (such as when GICs were larger than present or their equilibrium–line altitude (ELA) over time), however to summarize across all records, we use the most common and

robust evidence–glacier presence versus absence (or smaller than present) in each watershed. We calculated the percent of GICs that were absent or smaller than present in 100–year intervals from 12–0 ka for each region, and Arctic–wide (**Fig. 3–10**). Given that the resolutions of the chronologies are typically greater than the response time of a small mountain glacier or ice cap to a climate perturbation, we follow Solomina et al. (2015) and do not adjust for GIC response time as it can be considered negligible relative to the uncertainty of most reported ages. We round the

timing of all reported GIC evidence to the nearest 100–year interval in our analyses. In addition to documenting the timing of GIC variations, and to further investigate geographic patterns and controls of glacier fluctuations, we measured GIC elevation and the approximate steady–state ELA using geospatial methods for each study location. For the ELA estimates, we utilized a toolbox developed for automated calculation of GIC ELAs (Pellitero et al., 2015). We used the Accumulation–area ratio method (AAR) with an assumed AAR value of 0.67 (a common value for high–

latitude mountain glaciers in equilibrium) (Gross et al., 1976; Braithwaite and Muller, 1980). We note that the current extents of the GICs are likely not in equilibrium with present climate, and thus the actual ELA is likely higher in elevation in most cases. Present–day GIC extents were derived from the Global Land Ice Measurements from Space (GLIMS) dataset (Raup et al., 2007) and the ArcticDEM 32–meter resolution mosaic product (Porter et al., 2018) was used to derive all elevation measurements.


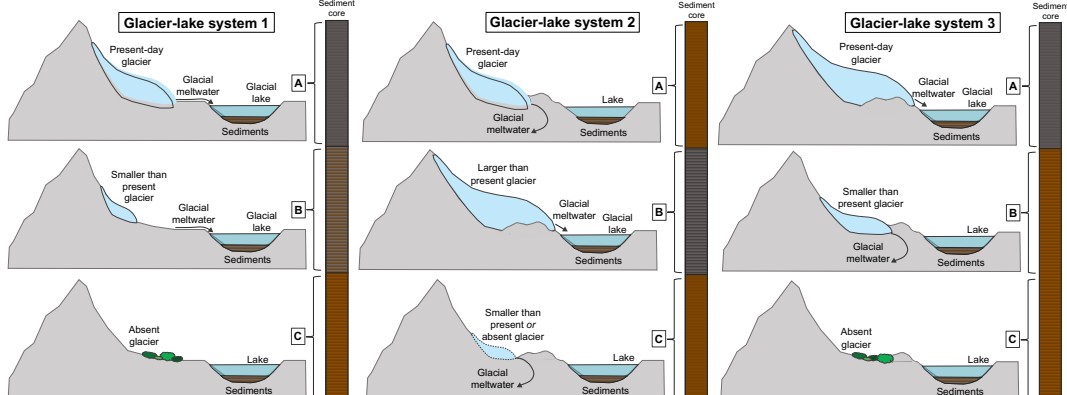

**Figure 2: Schematic of common glacier–lake systems and resulting lake sediment stratigraphies.** Left panel: Entire glacier resides within the lake's catchment and glacial meltwater enters the lake at present. Lake sediment records glacier presence (A:



fine–grained, gray minerogenic–rich sediment), absence (C: brown organic–rich sediment), and potentially more subtle variations
in glacier size over time (e.g., B: an intermediate sediment type). Middle and right panels: Classic "threshold lake" settings where
overtopping of/retreat behind a topographic threshold creates an "on/off" signal. In system type–2, the glacier is not currently in
the lake's catchment and glacial meltwater does not enter the lake at present. Lake sediments can record when the glacier was larger
than present and crossed the topographic threshold (B: fine–grained, gray minerogenic–rich sediment), and when the glacier was
at its present–day size, smaller, or absent and thus behind the topographic threshold (A and C: brown organic–rich sediment). In
this setting, whether a glacier was at present–day size (2A), smaller than present or absent (2C) may be indistinguishable from the
sediment record. In system type–3, glacial meltwater enters the lake at present. Lake sediments record glacier presence at present–
day size (A: fine–grained, gray minerogenic–rich sediment), and when the glacier was either absent or smaller than present and
retracted behind the topographic threshold (B and C: brown organic–rich sediment). In this setting, retracted glacier size (3B) may
be indistinguishable in the lake sediment record from glacier absence (3C).

## 3 Regional and pan–Arctic compilations of Holocene GIC records

### 3.1 Alaska

Alaska is located in northwest North America and is bounded by the Gulf of Alaska to the south, the Bering Sea to
the west, and the Arctic Ocean to the north. The climate of Alaska varies from maritime along the southern coast, to
transitional/continental in the interior, and Arctic in the north. GICs cover about 75,000 km$^2$, or ~5% of Alaska
(Calkin, 1988). The most extensive ice complexes are in the south along the margins, peripheral to the Gulf of Alaska,
the main source of precipitation to the region (Calkin, 1988; Kaufman and Manley, 2004; Barclay et al., 2009).

Six lake–based Holocene GIC records are available from south–central and southwest Alaska (**Fig. 3**). McKay and
Kaufman (2009) present two lake records in the Chugach Mountains area, in south–central Alaska (**Fig. 3**, records 1
and 2). The record from Greyling lake (**Fig. 3**, record 1) suggests general warmth, and diminished, if not entirely
ablated, glaciers between ~10–6 ka, and glacier regrowth in the catchment at ~4 ka (McKay and Kaufman, 2009). The
record from Hallet lake (**Fig. 3**, record 2) suggests smaller than present or absent glaciers from at least ~7.8 ka (the
base of the core) to ~6 ka and glacier regrowth ~4.5 ka (McKay and Kaufman, 2009). In the northeastern Ahklun
Mountains in southwestern Alaska, a record from Waskey lake (**Fig. 3**, record 3) indicates that glaciers lingered until
~9.1 ka, perhaps under conditions of abundant winter accumulation (Levy et al., 2004). Between ~9.1–3.1 ka glaciers
were less extensive than present and may have melted away entirely. Glaciers were reactivated and increased their
meltwater discharge into the lake at ~3.1 ka (Levy et al., 2004).

The following three records are from threshold lakes that do not receive glacial meltwater at present, so that glacial
meltwater input is indicative of glaciers more extensive than today. Zander et al. (2013) present a Holocene record of
Sheridan glacier from Cabin lake at the base of the Chugach Mountains (**Fig. 3,** record 4). The record indicates that
meltwater from Sheridan glacier entered the lake between roughly ~11.2–11 ka, at ~0.8 ka, from ~0.7–0.4 ka, and
from ~0.3–0.2 ka (Zander et al., 2013). In southern Alaska, in the Kenai Mountains, a record from Emerald lake
indicates that Grewingk glacier overtopped the topographic divide and delivered glacial sediments to the lake between





~10.7–9.8 ka and from ~0.6–0.2 ka, except for between ~0.5–0.4 ka when organic rich sediment was deposited indicating that the glacier was behind the divide for a short time (LaBrecque and Kaufman, 2016; **Fig. 3**, record 5).

Finally, a record from Goat lake suggests that North Goat glacier thickened to the threshold of the basin and discharged meltwater directly into the lake from ~10.6 to 9.5 ka, and between ~0.3–0.1 ka (Daigle and Kaufman, 2008; **Fig. 3**, record 6).

In summary, GICs first became smaller or absent altogether in Alaska in the early–to–middle Holocene, between ~10–

7.8 ka. Combined, the available lake–based records from Alaska suggest that 100% of GICs were smaller than present or absent between ~9.1 and 4.5 ka. GICs regrew between ~4.5 and 3.1 ka. A review of Holocene glacier fluctuations in Alaska suggests that land–based GICs were retracted during the early to middle Holocene, and that Neoglaciation began in some areas by ~4 ka, with more major advances by ~3 ka (Barclay et al., 2009).

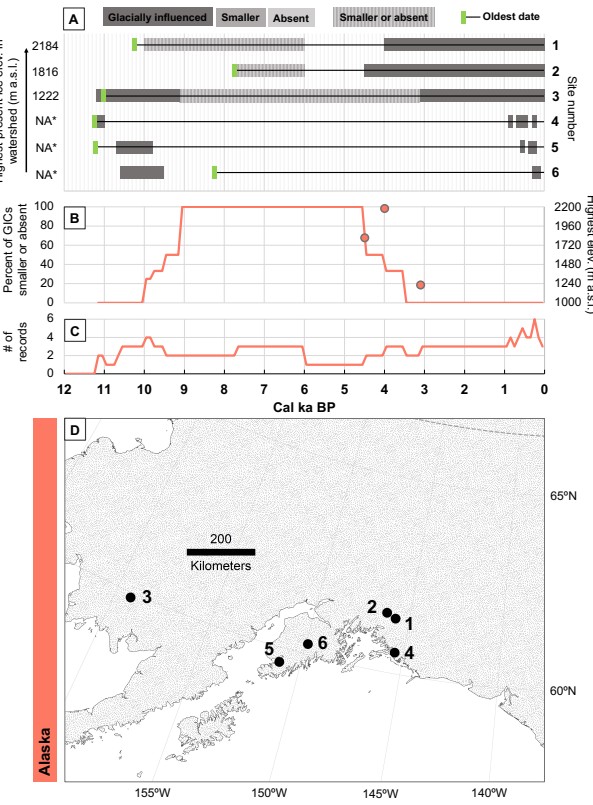


**Figure 3: Holocene lake–based GIC records from Alaska. A.** Shaded bars show a schematic view of GIC fluctuations reconstructed from lake sediments between 12–0 ka. Dark gray bars denote glacially influenced sediment, medium gray bars denote GICs smaller than present but not absent, and light gray bars denote GIC absence. Striped bars denote GICs smaller than present *or* absent (where these two states cannot be distinguished from one another). Colored bars indicate the oldest reported age in each

record (green=[14]C–dated plant or aquatic macrofossil, yellow=paleomagnetic secular variation (PSV), blue=[14]C dated marine



macrofossil, pink=bulk sediment, purple=tephra, black=other). Values on the left show each watershed's highest present ice elevation in meters above sea level (m a.s.l.). **B.** Line shows the percent of this region's studied GICs smaller than present or absent from 12 to 0 ka. Open dots show the timing of earliest GIC regrowth and solid dots the timing of sustained GIC regrowth in the middle–to–late Holocene, versus each watershed's highest present ice elevation. **C.** The number of records from this region from 12 to 0 ka, and **D.** study locations. Sites are: **1.** McKay and Kaufman, 2009 (Greyling lake/unnamed glaciers); 2. McKay and Kaufman, 2009 (Hallet lake/unnamed glaciers); **3.** Levy et al., 2004 (Waskey lake/unnamed glaciers); **4.** Zander et al., 2013 (Cabin lake/Sheridan glacier); **5.** LaBrecque and Kaufman, 2016 (Emerald lake/Grewingk glacier); **6.** Daigle and Kaufman, 2008 (Goat lake/North Goat glacier). We note that the oldest radiocarbon age from Greyling lake is pre–Holocene. We report the oldest Holocene age. *Lake records 4, 5, & 6 are considered threshold records and do not receive glacial meltwater at present. Thus, the non–glacial/glacial transitions may not precisely represent the timing of the ice cap's response to climate forcings.

### 3.2 Arctic Canada (Baffin Island, northeast Canada)

The Canadian Arctic Archipelago lies north of Canada's continental mainland and is bounded in the east by Baffin Bay and the Davis Strait, and in the north by the Arctic Ocean. The climate of the archipelago's largest island, Baffin Island, is influenced by the interplay between the cold, low–salinity Baffin Current running south along the western side of Baffin Bay, and the relatively warm subarctic waters of the West Greenland Current, which transport heat to the Baffin Bay region affecting sea–ice conditions, as well as terrestrial climate. Mean annual temperatures range from −15°C in northern Baffin Island, to −5°C in the south, while mean July temperatures are ~4°C in coastal areas and generally warmer inland (ESWG, 1995). Precipitation is generally ~200–300 mm annually (ESWG, 1995). The island currently hosts the Barnes Ice Cap in central Baffin Island, and Penny Ice Cap, located ~300 km south, as well as numerous small mountain GICs located along the eastern mountains.

Five lake–based records of Holocene GIC variability are available from Arctic Canada, all of which are located on Baffin Island. On northeastern Baffin Island, Thomas et al. (2010) present Holocene records from two proglacial lakes (Longspur and Big Round lake; **Fig. 4**, records 1 and 3) and two threshold lakes (Yougloo and Igloo Door lake; **Fig. 4**, records 4 and 5). The Big Round lake record (**Fig. 4**, record 3) suggests that Kuuktannaq glacier was present in the lake's catchment from at least ~10–6 ka, and from ~2 ka to the present. Minimum glacier extent is reported between ~6 and 2 ka (Thomas et al., 2010). The record from Igloo Door lake (**Fig. 4**, record 5) suggests that Kuuktannaq glacier did not cross the topographic threshold, causing silt–laden glacial meltwater to be diverted into the lake basin, until ~1.1 ka. Similarly, the Yougloo lake record (**Fig. 4**, record 4) suggests very little glacial input throughout the Holocene, until Kuuktannaq glacier advanced across a topographic threshold sometime after ~1.7 ka (Thomas et al., 2010). Thus, the late Holocene advance of Kuuktannaq glacier was the most extensive since at least ~10.2–10.1 ka (the basal ages from Yougloo and Igloo Dooor lakes). Finally, the record from proglacial lake, Longspur (**Fig. 4**, record 1), which at present receives meltwater from five glaciers that terminate 1 km or less from the lake, suggests minerogenic sediment input throughout the Holocene, implying that alpine glaciers persisted from at least ~9.2 ka (based on the age model from the shallow core site) (Thomas et al., 2010). If or when the glaciers were smaller is not reported. On Cumberland Peninsula, the easternmost point on Baffin Island, a Holocene record of Caribou glacier from Donard lake shows





organic–rich gyttja deposition until ~9.5 ka, clastic laminae between ~9.5–8.6 ka, a return to organic–rich
sedimentation between ~8.6–5.7 ka, and clastic deposition from ~5.7 ka to present (Moore et al., 2001; Miller et al.,
2005; **Fig. 4**, record 2). Glacially derived sediments from this record indicate time periods when Caribou glacier had
thickened enough to breach the col separating it from the valley of Donard lake, which greatly increased the lake's
catchment size to cause delivery of glacial sediment (Moore et al., 2001; Miller et al., 2005). We exclude two lake
records: lake Jake from Miller et al. (2005) and Tasikutaaq lake from Lemmen et al. (1988) from our summary due to
unclear interpretations and/or uncertain chronologies.

In summary, GICs first became smaller or absent altogether on Baffin Island in the early–to–middle Holocene,
between ~10.2–6 ka. At least 80% of the lake–based records from Arctic Canada indicate that GICs were smaller than
present or absent between ~10.2–10 ka and ~5.9–5.7 ka, and at least 60% were smaller than present or absent between
~10.2–9.5 ka and between ~8.6–2 ka. The earliest GIC regrowth occurred in southern Baffin Island at ~5.7 ka on
peaks roughly ~1400 m a.s.l., and between ~2 and 1.1 ka in the northeast on peaks ~1150 m a.s.l. and below. A review
of latest Pleistocene and Holocene glaciation of Baffin Island suggests that at least some alpine GICs survived the
warmer than present early HTM, and that GICs advanced beginning in some places as early as ~6 ka (although most
do not record near–LIA positions until ~3.5–2.5 ka) (Briner et al., 2009).


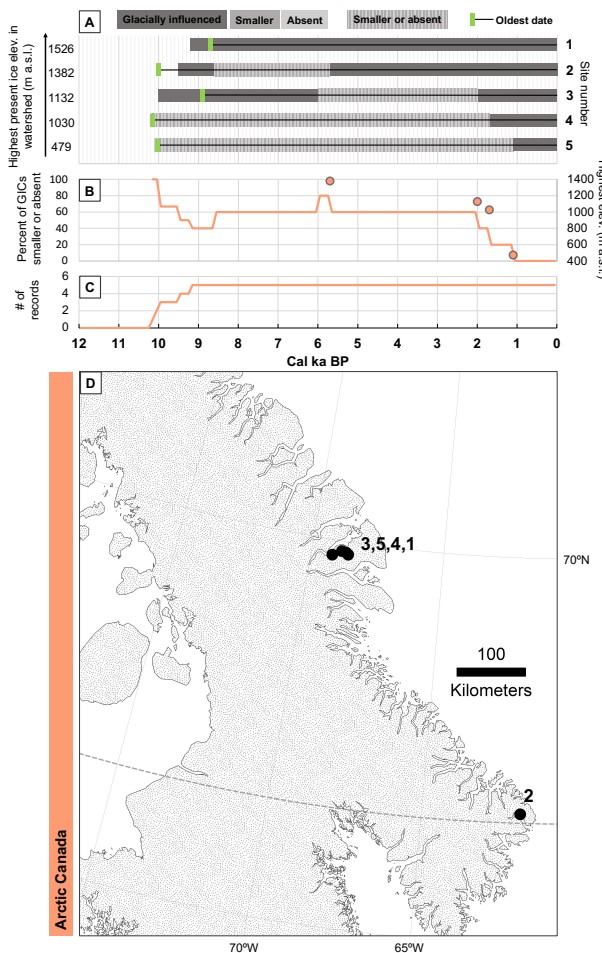

**Figure 4: Holocene lake–based GIC records from Arctic Canada (Baffin Island).** Panel descriptions, symbols and colors are the same as in Figure 3. Sites are: **1.** Thomas et al., 2010 (Longspur lake/unnamed glaciers); **2.** Moore et al., 2001 & Miller et al., 2005 (Donard lake/Caribou glacier); **3.** Thomas et al., 2010 (Big Round lake/Kuuktannaq glacier); **4.** Thomas et al., 2010 (Yougloo lake/Kuuktannaq glacier); **5.** Thomas et al., 2010 (Igloo Door lake/ Kuuktannaq glacier). We note that the oldest radiocarbon age from Donard lake is pre–Holocene. We report the oldest Holocene age.

### 3.3 Greenland

Greenland is over 2 million km$^2$ in area, extending from ~59 to 83°N between the Atlantic and Arctic Oceans. GICs cover almost 90,000 km$^2$ of the land outlying the Greenland Ice Sheet (GrIS). The periphery of the vast island hosts a wide range of modern climates and is highly influenced by various ocean and atmospheric processes, sea ice extent, and the presence of the GrIS. In Nuuk (Greenland's capitol in the southwest), average summer air temperature is ~5.8°C, and average annual air temperature is –1.4°C. For contrast, in Thule (in northwest Greenland) summer air



temperature is ~4.2°C and annual air temperature is –10.9°C (Cappelen, 2020; reported air temperature is the 1981–
2010 average). Precipitation is generally higher on the coasts than inland and is especially high in the very south
(reaching up to ~3000 mm per year) and on the eastern coast, and low in the north (in Thule annual accumulated
precipitation is ~132 mm) (Cappelen et al., 2001; Cappelen, 2020).

Twenty–two lake–based records of Holocene GIC variability are available from Greenland. Beginning in the south
near Kap Farvel, a record from Quvnerit lake suggests that glaciers were present in the catchment in the early Holocene
and into the mid–Holocene, between at least ~9.5 to 7.1 ka (Larocca et al., 2020a; **Fig. 5**, record 14). From ~7.1–5.5
ka glaciers were absent from the lake's catchment and likely melted away entirely. Between ~5.5–3.1 glaciers were
smaller than present or absent. Persistent glacial input into the lake occurred from ~3.1 ka to present (Larocca et al.,
2020a). Roughly ~95 km to the northwest, a record from Uunartoq lake suggests that glaciers likely melted away
completely sometime prior to ~5.2 ka and remained absent from the catchment until they regrew ~1.2 ka (Larocca et
al., 2020a; **Fig. 5**, record 4). Less than 20 km NW, the record from Alakariqssoq lake suggests that glaciers were
present in the lake catchment from at least ~10.75 to ~7.3 ka (Larocca et al., 2020a; **Fig. 5**, record 3). Glaciers were
likely absent all together between ~7.3 and ~1.3 ka. Sustained regrowth of glaciers in Alakariqssoq's catchment
occurred at ~1.3 ka (Larocca et a., 2020a).

On Ammassalik Island, ~100 km south of the Arctic Circle on the coast of southeast Greenland, van der Bilt et al.
(2018) present a Holocene record of Ymer glacier from Ymer lake. The record indicates the onset of lake
sedimentation at ~10 ka, and that the lake catchment remained glaciated until ~9.5 ka (**Fig. 5**, record 17). Cessation
of glacial input occured at ~9.5 ka and glaciers are reported to have reformed in the catchment after ~1.2 ka (van der
Bilt et al., 2018). Just NW of Ymer lake, a new proglacial lake record from Smaragd Sø indicates that Mittivakkat
Glacier was probably smaller than present from at least ~7.9 ka (and possibly as early as ~11.4 ka) (Larsen et al.,
2021; **Fig. 5**, record 22). Radiocarbon dating of dead plants and reindeer antlers indicate that the glacier began to
expand sometime between ~1.4 and 0.7 ka, when Mittivakkat glacier had advanced to its LIA extent. Aerial
photographs show that the glacier retreated out of the catchment again prior to 1933 CE (Larsen et al., 2021). However,
since Smaragd Sø is glacially fed only when the Mittivakkat glacier is at an advanced position, the lake record is not
optimal for recording Holocene glacier variations and can only be used to determine when the glacier was at a
maximum position (Larsen et al., 2021). Nearby, a 9.5 ka record from Kulusuk lake on Kulusuk Island, suggests that
glaciers delivered meltwater to the lake until ~8.7 ka (Balascio et al., 2015; **Fig. 5**, record 21). At ~8.7 ka, significant
retreat of the Kulusuk glaciers is reported, which was interrupted by two glacier advances at ~8.5 and ~8.2 ka. Between
~7.8 and 4.1 ka, high organic content implies that the glaciers likely melted away completely. From ~4.1 ka, a series
of episodic advances followed by periods of retreat occurred, which were superimposed on a gradual trend toward
larger glacier size. After ~1.3 ka, the Kulusuk glaciers stabilized and reached a greater size (Balascio et al., 2015). At
a similar latitude, but on Greenland's southwest coast, sediments from proglacial Crash lake suggest that after an
advance at ~9 ka, glaciers experienced net recession until ~4.6 ka (Schweinsberg et al., 2018; **Fig. 5**, record 13). The



onset of the Neoglacial is recorded in the Crash lake sediments ~4.6 ka, followed by increasing glacier size, as well as several intervals of glacier advances.

Near Nuuk, Larsen et al. (2017) present records of Holocene GIC variability from three lakes fed by GICs with different elevation ranges (**Fig. 5**, records 8, 11, and 18). The Badesø lake record (**Fig. 5**, record 8), which today is fed by the highest altitude glaciers, indicates that after the lake's isolation from the sea at ~8.5 ka, glacier ice was absent from the catchment until ~5.5 ka. Renewed ice growth and meltwater influx began again at ~5.5 ka and continued to present, except for a short period of glacier absence between ~4.4–3.5 ka (Larsen et al., 2017). The record from Langesø lake (**Fig. 5**, record 11) suggests that from the lake's isolation at ~8.7 ka and until 3.6 ka, organic–rich gyttja was deposited, reflecting that all glacier ice had melted in the catchment. Glacier regrowth began at ~3.6 ka and meltwater input continued to present (Larsen et al., 2017). The record from lake IS21 (i.e., **Fig. 5**, record 18), which is currently fed by the relatively low elevation Qasigiannguit ice cap, indicates that the lake formed before ~9 ka, and that glacial meltwater input occurred until ~7.9 ka. Between ~7.9–1.6 ka, the ice cap had completely melted away. Renewed ice growth began at ~1.6 ka, and a period of reduced meltwater input is reported between ~1.4–0.8 ka (Larsen et al., 2017). Around 50 km south of Nuuk, near Buksefjord, a record from Pers lake suggests that from the time the lake emerged from sea at ~8.6 ka, glaciers were absent from the catchment, until ~4.3 ka (Larocca et al., 2020b; **Fig. 5**, record 16). After ~4.3 ka the GICs were either smaller than present or completely absent at times until ~1.4 ka, at which time the GICs persistently remained in the catchment until present (Larocca et al., 2020b). We exclude a second lake record (lake T3) from this study from our summary due to poor age constraints. However, the record suggests that following emergence of lake T3 between ~8.4–7.5 ka, the lake received continuous glacial meltwater input (although reduced for an extended period of unknown age and duration) through the remainder of the Holocene, implying that some high elevation GICs may have survived the HTM.

On the peninsula of Liverpool Land in central east Greenland, Lowell et al. (2013) present a Holocene record of Istorvet ice cap from Bone lake (**Fig. 5**, record 12). The record indicates that Istorvet ice cap did not feed meltwater into the lake throughout most of the Holocene (between ~9.7–0.8 ka). Since the lake does not receive glacial input today, it can only be concluded that the ice cap was at a size similar to or smaller than at present, however the study suggests that during the middle and late Holocene, Istorvet ice cap was likely small. Radiocarbon dates from subfossil plants provide strong evidence that the ice cap was smaller than present from AD 200 to 1025. Following this, the most extensive advance during the Holocene occurred from AD 1150 to at least AD 1660 (Lowell et al, 2013). In the Scoresby Sund region of central east Greenland, Levy et al. (2014) report a record of Bregne ice cap from Two Move lake (**Fig. 5**, record 15). Between ~10–2.6 ka, the lake environment was dominated by biological production implying no glaciogenic input. The regrowth of Bregne ice cap during the late Holocene is first reported at ~2.6 ka, while sustained ice expansion is reported at ~1.9 ka, after which a late Holocene maximum was reached by ~0.74 ka (Levy et al., 2014). Medford et al. (2021) present a ~12 ka record of SW Renland Ice Cap from Rapids and Bunny lakes (**Fig. 5**, record 1). The lake records suggest that deglaciation began as early as ~12.7 ka, and that by ~9.5 ka the ice cap had retreated behind its present–day extent. The ice cap remained smaller than present during most of the



Holocene, however periodic inputs of inorganic sediments to the lakes suggest repeated fluctuations of Renland Ice Cap, particularly between ~7.6–7.2 ka and ~3.4–3.2 ka, the onset of Neoglaciation. A brief episode of glacier expansion is noted at ~1.3 ka, followed by the onset of significant late Holocene glaciation shortly after ~1.05 ka.


Schweinsberg et al. (2017, 2019) present three Holocene records of glacier variability on Nuussuaq, West Greenland (**Fig. 5**; records 5, 9, and 10). The Pauiaivik lake record suggests that Sermikassak glacier was smaller or absent between ~9.5–4.3 ka (Schweinsberg et al., 2019; **Fig. 5**, record 5). Mineral–rich sediments are reported throughout the last ~4.3 ka, implying glacier presence in the lake's catchment (Schweinsberg et al., 2019). The record from Saqqap Tasersua lake (**Fig. 5**, record 9) suggests that Qangattaq ice cap was active in the catchment until roughly ~10.2 ka (Schweinsberg et al., 2019). Between ~10.2–4.5 ka, the record suggests that the ice cap was mostly either reduced or not active in the catchment. However, some glacier activity is reported before ~8.5 ka, though the ice cap's relative size is not reported. Periods of enhanced glacier activity are reported between ~8.5–8.2, ~7.2–6.8, and ~5.6–5.3 ka. There is a gap in the downcore proxy data, however enhanced glacier activity commenced again in the late Holocene, sometime between ~3 and 2 ka (Schweinsberg et al., 2019). We note that we report interpretations from Fig. 10 in Schweinsberg et al. (2019). The Sikuiui lake record (**Fig. 5**, record 10) suggests that Qangattaq ice cap was either reduced or not active in the catchment between ~9.4–3.8 ka, except for mineral–rich units between ~8.8–8 ka and around ~5.7 ka, that may represent brief glacier advances. The onset of the Neoglacial and regrowth of the ice cap is reported at ~5 ka, with more substantial snowline lowering and expansion at ~3.7 ka, followed by other expansion phases at ~2.9, 1.7, and 1.4 ka, and during the LIA (Schweinsberg et al., 2017; 2019).




In northeast Greenland, a late Holocene record of Slettebreen ice cap from Madsen lake suggests that the ice cap was present in the catchment since at least ~1.8 ka (Adamson et al., 2018; **Fig. 5**, record 6). In northwest Greenland, a record from Deltasø lake indicates that North ice cap was smaller than present or absent through most of the Holocene, from at least ~10.1 to ~1850 AD when the ice cap reached its present–day size (Axford et al., 2019; **Fig. 5**, record 7). Finally, in Finderup Land, north Greenland, Larsen et al. (2019) present five proglacial lake records of Holocene GIC activity that demonstrate that GICs in Finderup Land survived the HTM, possibly due to increased precipitation via a reduction in sea ice extent and/or increased poleward moisture transport. The record from lakes T3 and T8 (**Fig. 5**, record 2) suggest that Flade Isblink ice cap survived the HTM but was smaller than present between ~9.4–0.2 ka (Larsen et al., 2019). The record from lake T4 indicates that Ice Cap 1 delivered glacial meltwater to the lake from at least ~5.9 ka, the estimated time of the lake's isolation (Larsen et al., 2019; **Fig. 5**, record 19). Likewise, the lake records from T2 and T6 show that Ice Cap 2 delivered meltwater through the Holocene, from at least ~9.5 ka, the estimated time of isolation (Larsen et al., 2019; **Fig 5**, record 20).




In summary, GICs first became smaller or absent altogether in Greenland in the early–to–middle Holocene, between ~10.2–7.1 ka. Combined, of the available lake–based records from Greenland, at least 75% suggest that GICs were smaller than present or absent between ~8–3.7 ka. The highest percentage (~94%) of GICs either smaller than present or absent occurred in the middle Holocene between ~6.8–5.9 ka. Glaciers first began to regrow in lake catchments



after ~5.7 ka. The largest decreases in the percentage of smaller or absent GICs occurred between roughly ~4–3 ka and especially between ~2–1 ka. A review of the fluctuations of local glaciers during latest Pleistocene and Holocene suggests that during the HTM, most local glaciers in Greenland were smaller than at present and may have disappeared completely, and that generally GICs grew to their maximum Holocene extents during historical time (Kelly and Lowell, 2009).




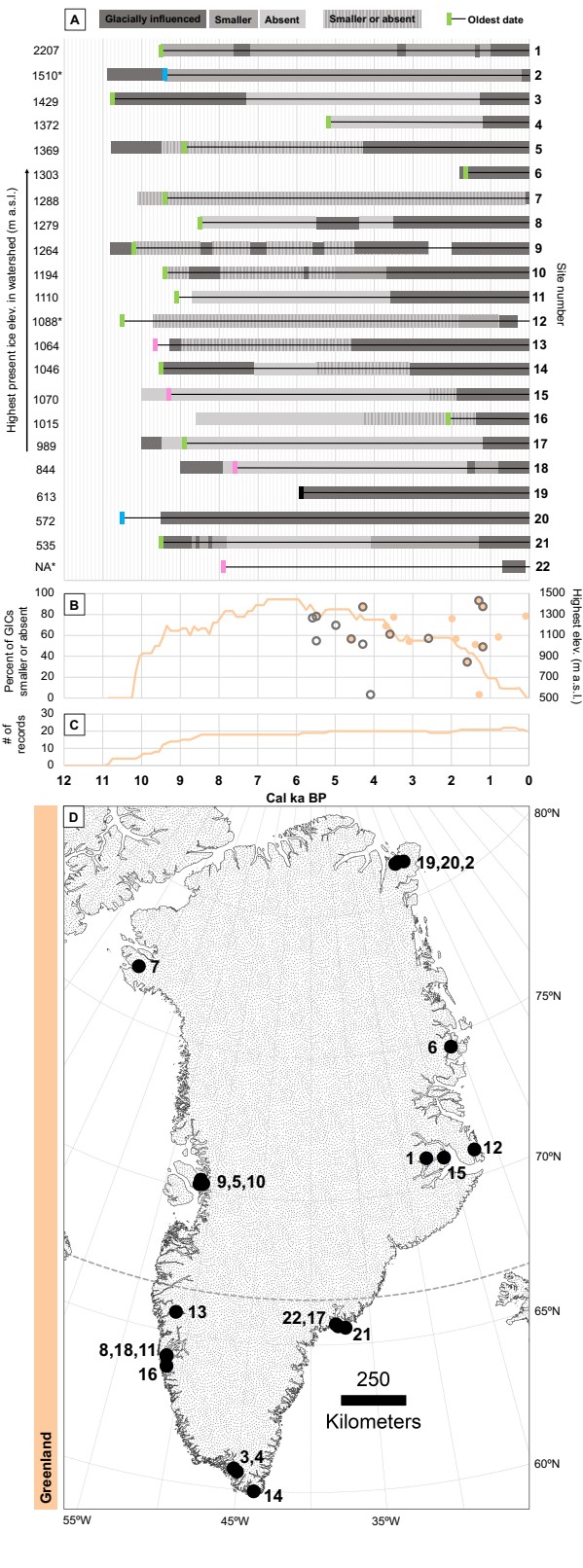



**Figure 5: Holocene lake–based GIC records from Greenland.** Panel descriptions, symbols and colors are the same as in Figure 3. Sites are: **1.** Medford et al., 2021 (Rapids and Bunny lakes/Renland Ice Cap); **2.** Larsen et al., 2019 (lake T3 and T8/Flade Isblink ice cap); **3.** Larocca et al., 2020a (Alakariqssoq lake/unnamed glaciers); **4.** Larocca et al., 2020a (Uunartoq lake/unnamed glaciers);

**5.** Schweinsberg et al., 2019 (Pauiaivik lake/Sermikassak glacier); **6.** Adamson et al., 2018 (Madsen lake/Slettebreen ice cap); **7.** Axford et al., 2019 (Deltasø lake/North ice cap); **8.** Larsen et al., 2017 (Badesø lake/RGI40–05.07196 and RGI40–05.07208 glaciers); **9.** Schweinsberg et al., 2019 (Saqqap Tasersua lake/Qangattaq ice cap); **10.** Schweinsberg et al., 2017 (Sikuiui lake/Qangattaq ice cap); **11.** Larsen et al., 2017 (Langesø lake/RGI40–05.07196 glacier); **12.** Lowell et al., 2013 (Bone lake/Istorvet ice cap); **13.** Schweinsberg et al., 2018 (Crash lake/unnamed glacier); **14.** Larocca et al., 2020a (Quvnerit lake/unnamed glaciers);

**15.** Levy et al., 2014 (Two Move lake/Bregne ice cap); **16.** Larocca et al., 2020b (Pers lake/unnamed glaciers); **17.** van der Bilt et al., 2018 (Ymer lake/Ymer glacier); **18.** Larsen et al., 2017 (lake IS21/Qasigiannguit ice cap); **19.** Larsen et al., 2019 (lake T4/Ice cap 1); **20.** Larsen et al., 2019 (lake T2 and T6/Ice cap 2); **21.** Balascio et al., 2015 (Kulusuk lake/ Kulusuk glaciers); **22.** Larsen et al., 2021 (Smaragd Sø/Mittivakkat Glacier). *We report the highest present ice elevation of Flade Isblink ice cap and Istorvet ice cap (i.e., study 2 and 12) rather than the highest ice elevation inside the lake's watersheds. We also note that study 12 and 22 are

considered threshold records and the lakes does not receive meltwater from the glacier or ice cap at present, and that the oldest reported date from study 12 is from a nearby lake, Snowbank lake. Study 6 (i.e., Adamson et al., 2018) is a short record and only records a minimum bound on glacier regrowth in the late Holocene. Schweinsberg et al. (2017) reports that the thin minerogenic rich layer seen in the Sikuiui lake record at ~5.7 ka "may represent a brief glacier advance." Thus, because of this uncertainty, we report the earliest regrowth of the Qangattaq ice cap at ~5 ka BP. Finally, we note that the oldest age from lake T4 (i.e., study 19;

Larsen et al., 2019) is an isolation age based on correlation between the lake's elevation and a local relative sea–level curve.

**3.4 Iceland**

Located just below the Arctic circle, Iceland sits in the North Atlantic, and is situated at the border between warm and

cold ocean currents, creating a steep climate gradient across the island (Einarsson, 1984; Geirsdóttir et al., 2009). Iceland's climate is maritime with cool summers and mild winters (Einarsson, 1984). It is moderated by the Irminger current, composed of warm, saline waters which wrap around the island's southwest coast, as well as cold and low salinity waters via a branch of the East Greenland Current, which flows in a southeasterly direction around Iceland's northern coast (Björnsson and Pálsson, 2008). About 11% of the ~100,000 km$^2$ island is covered by glaciers and the

largest ice caps sit in the southern and central highlands (Björnsson and Pálsson, 2008). Mean temperature is generally close to 0°C in the winter and is ~10°C in the summer (Furger et al., 2007). Precipitation is generally controlled by both orography and prevailing winds, and annually is highest on Iceland's south and southeast coast, and lowest in the inland northern regions (Björnsson and Pálsson, 2008; Anderson et al., 2019).

In Iceland, five lake–based records of Holocene GIC fluctuations are available (**Fig. 6**). Three records lie in the Westfjords region. The northernmost record (i.e., Harning et al., 2016a; **Fig. 6**, record 5) from Skorarvatn lake shows that the northern margin of Drangajökull ice cap reached a size comparable to its present limit by ~10.3 ka. The ice cap is interpreted to have been ~20% smaller than present by ~9.2 ka, and likely melted away by ~9 ka, as local peak warmth is reported between ~9 and 6.9 ka (Harning et al., 2016a). A record of SE Drangajökull ice cap from

Tröllkonuvatn lake (i.e., Harning et al., 2016a, 2016b; **Fig. 6**, record 3) suggests that the ice cap was present in the





lake's catchment between ~10.3–8.75 ka and was absent between ~8.75–1 ka. From ~1 ka, the ice cap was present, except for a short non–glacial interval, between ~0.7–0.55 ka (Harning et al., 2016a, 2016b). From the same two studies, a record from Efra–Eyvindarfjarðarvatn lake (i.e., Harning et al., 2016a, 2016b; **Fig. 6**, record 4) suggests that SE Drangajökull ice cap was present in the lake's catchment between ~10.3–9.2 ka, and was absent from at least ~9–

2.3 ka. After ~2.3 ka, the ice cap remained in the lake's catchment, except for a short non–glacial interval between ~1.5–1.4 ka (Harning et al., 2016a, 2016b). Additional evidence of the Holocene history of Drangajökull ice cap from seven threshold lakes is presented by Schomacker et al. (2016). We exclude these records from our summary because six of the lakes do not receive meltwater from the ice cap today making it difficult to constrain when the ice cap was smaller than present, and the record from the seventh, lake Skeifuvatn, which does receive meltwater from

Drangajokull's south margin, did not contain any dateable material. However, in contrast to the previously mentioned studies (i.e., **Fig. 6**, records 3, 4, & 5), the main conclusion is that the Drangajokull ice cap probably survived the HTM and was present through the entire Holocene, perhaps due to increased winter precipitation (Schomacker et al., 2016).

In south–central Iceland, two studies from lake Hvítárvatn, located on the eastern margin of Langjökull, the second largest ice cap in Iceland, show a nearly identical Holocene history (i.e., Black, 2008; Larsen et al., 2012). Evidence from diatom assemblages, as well as physical and chemical proxies suggest that Langjökull largely disappeared from at least ~10.2 to 8.7 ka, and from 7.35–5.5 ka (Black, 2008). The record suggests that Langjökull was then present in the catchment from ~5.5 ka (Black, 2008; we do not include this record in our compilation as it is unpublished).

Similarly, Larsen et al. (2012) report that following regional deglaciation, summer temperatures were already high enough that mountain GICs had melted away, and thus, no ice is reported in lake Hvítárvatn's catchment from ~10.2–8.7 ka (**Fig. 6**, record 2). The record suggests that this early Holocene warmth was interrupted by two pulses of cooling and possibly glacier growth between ~8.7–7.9 ka. Following this, ice–free conditions in the watershed and high within–lake productivity occurred during the HTM between ~7.9–5.5 ka. The inception and expansion of Langjökull

ice cap began at ~5.5 ka (Larsen et al., 2012). In eastern Iceland, a ~10.5 ka record from lake Lögurinn suggests that Eyjabakkajökull, a surge–type outlet glacier of the Vatnajökull ice cap, receded rapidly during the final phase of the last deglaciation (Striberger et al., 2012; **Fig. 6,** record 1). Glacial meltwater input ceased by ~9 ka and returned at ~4.4 ka, suggesting an almost 5 ka long glacier–free period during the early and mid–Holocene (Striberger et al., 2012).


In summary, GICs first became smaller or absent altogether in Iceland in the early Holocene, between ~10.2–8.75 ka. Between 80 to 100% of the lake–based records from Iceland indicate that GICs were smaller than present or absent from ~9–5.5 ka (**Fig. 6B**). In eastern and western Iceland, GICs regrew on relatively high elevation peaks between ~5.5 and 4.4 ka, and on the lower elevation peaks in the Westfjords region between ~2.3 and 1 ka (**Fig. 6A, B, and**

**D**). A review summarizing records of Holocene glacial and climate evolution suggests that by ~10.3 ka, the main ice sheet was in rapid retreat across the highlands of Iceland, and that the local HTM was reached after 8 ka, with land temperatures estimated to be ~3°C higher than the 1951–90 reference period, implying ice–free conditions in the early





to mid–Holocene (Geirsdóttir et al., 2009). The review also notes that many marine and lacustrine records indicate a substantial summer temperature depression between ~8.5–8 ka, and that the onset of Neoglacial cooling occurred after ~6 ka, with increased glacier activity between ~4.5–4 ka, which intensified further between ~3–2.5 ka (Geirsdóttir et al., 2009).

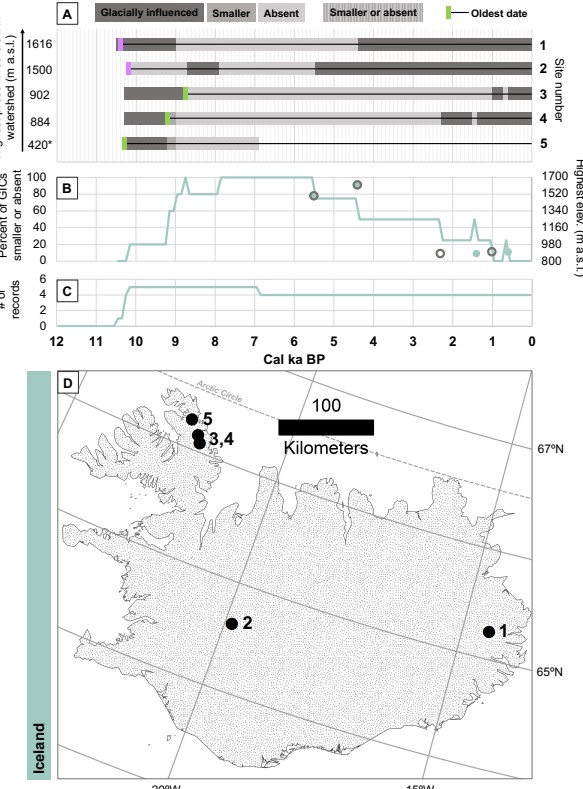

**Figure 6: Holocene lake–based GIC records from Iceland.** Panel descriptions, symbols and colors are the same as in Figure 3. Sites are: **1.** Striberger et al., 2012 (lake Lögurinn/Eyjabakkajökull glacier); **2.** Larsen et al., 2012 (lake Hvítárvatn/Langjökull ice cap); **3.** Harning et al., 2016a, 2016b (Tröllkonuvatn lake/Drangajökull ice cap**)**; **4.** Harning et al., 2016a, 2016b (Efra–Eyvindarfjarðarvatn lake/Drangajökull ice cap); **5.** Harning et al., 2016a (Skorarvatn lake/Drangajökull ice cap). We note that records 3, 4, and 5 are considered threshold lake records. Thus, the non–glacial/glacial transitions may not precisely represent the timing of the ice cap's response to climate forcings. *At present, Skorarvatn lake is outside of the Drangajökull ice cap's drainage area and 420 m a.s.l. is the reported catchment threshold for receiving glacial sediments.

### 3.5 Scandinavia

Located between ~57–71°N, the Scandinavian peninsula stretches over a large latitudinal area and hosts a range of modern climates. In general, the region's climate is heavily influenced by several oceanic and atmospheric processes,





and in particular by the advection of heat and moisture to the North Atlantic region, the position of the polar front, as well as the winter index of the North Atlantic Oscillation (Moros et al., 2004; Oien et al., 2020; Bakke et al., 2005c). Southern Scandinavia is temperate, with a strong west–to–east gradient from a maritime to continental climate, while northern Scandinavia is characterized by a subpolar–to–polar climate (Oien et al., 2020). Precipitation generally
decreases moving inland with distance from the coast, with the southern coastal areas receiving the highest amount of winter precipitation (Oien et al., 2020).

The Scandinavian peninsula is relatively data rich with twenty Holocene lake–based records of GIC variability available, covering a broad north–south transect in Norway and Sweden (**Fig. 7**). Moving from north to south, the
northernmost study (i.e., Wittmeier et al., 2015; **Fig. 7**, record 20) presents a record of glacier activity from the northern outlet of the Langfjordjøkelen ice cap via a chain of three downstream lakes. Following deglaciation of the valley of Sør–Tverrfjorddalen (~10 ka), the ice cap was reduced or absent until 4.1 ka, when Langfjordjøkelen ice cap reformed. An exception occurred at ~8.2 ka when several detrital parameters abruptly increased, possibly reflecting a reforming glacier (Wittmeier et al., 2015). Roughly 80 km southwest, pro–glacial lake Aspvatnet was isolated from the sea ~10.3
ka (i.e., Bakke et al., 2005a; **Fig. 7**, record 17). The lake record suggests that Lenangsbreene glacier was present between ~9.8 and 8.9 ka, and absent between ~8.8 and 3.8 ka. After 3.8 ka, the record shows continuous input of glacier–meltwater derived sediments to lake Aspvatnet indicating sustained glacier presence in the catchment (Bakke et al., 2005a). In northern Sweden, Snowball and Sandgren (1996) present a record derived from three lake basins (**Fig 7**, record 16) which suggest that Kårsa glacier likely disappeared during the early and mid–Holocene and
reformed ~3.3 ka. Just over 20 km southwest, a record from Vuolep Allakasjaure lake (i.e., Rosqvist et al., 2004; **Fig. 7**, record 18) suggests that the area was ice free and vegetated at ~9.7 ka, and that a glacier was present in the lake's catchment during the last five–thousand years. In northern Norway, sediment cores from distal–fed glacial lakes, Vestre– and Austre Kjennsvatnet (i.e., Bakke et al., 2010; **Fig. 7**, record 10), suggest that the ELA of Austre Okstindbreen glacier was at its highest during the entire Holocene between ~7–4.9 ka, but possibly survived the HTM.
Expanded ice cover is reported in the catchment from ~4.9 ka, but the glacier was generally small from ~3.95 until 1.3 ka (Bakke et al., 2010). We exclude the late Holocene portion of this record because the precise timing of when the glacier was smaller than at present is unclear.

In southern Norway, a group of twelve lake–based GIC records are located in the vicinity of Jostedalsbreen, the largest
glacier in mainland Europe. Moving from the coast, inland, a late Holocene record of glacier activity from lake Grøndalsvatn (i.e., Nesje et al., 1995; **Fig. 7**, record 19) suggests that Ålfotbreen glacier briefly expanded between ~2.7–2.1 ka. Minor glacier activity in the catchment is reported from ~1.5 ka, however the main increase in Neoglacial activity occurred after ~0.8 ka (Nesje et al., 1995). A record from distal glacier–fed lake, Sørsendalsvatn, shows that Blåbreen glacier reached a minimum size ~9 ka, advanced at ~8.4–8.2 ka, then abruptly decreased in size after ~8.2
ka (Bakke et al., 2013; **Fig. 7**, record 15). Between ~8 and 5.5 ka low or no glacial activity is reported. From ~5.7 to 2 ka uncertainties in the dating and differences between the core records did not allow for the detailed interpretation of glacier activity, thus, we exclude much of this section of the record. The onset of the Neoglacial is reported at ~5.5



ka, as there was a major change in the sedimentation regime at that time. Increased glacier activity in Sørsendalsvatn's catchment is reported between ~2–0.8 ka, and glacier extent is reported as largest during the last 1000 years, especially

during the Little Ice Age (Bakke et al., 2013). A record from Grøningstølsvatnet lake indicates that the catchment deglaciated ~9.5 ka, and that the Grovabreen glacier was absent until ~4.7 ka, except for a glacier episode between ~8.4–7.9 ka (Seierstad et al., 2002; **Fig. 7**, record 12). Grovabreen glacier has existed continuously from ~4.7 ka (Seierstad et al., 2002). Nesje et al. (2001) present a Holocene record of Flatebreen glacier from lake Jarbuvatnet. The record indicates a glacier expansion episode which terminated ~10.2 ka, a second major glacier phase from ~8.4–8.1

ka, and two periods with little or no glacier activity in the lake catchment from ~10–8.4 ka and ~8.1–4 ka (**Fig. 7**, record 13). Flatebreen was present in the catchment from ~4 ka to present (Nesje et al., 2001). Vasskog et al. (2012) present a sedimentary record from lake Nerfloen. The lake's large catchment encompasses ~440 km$^2$ and presently hosts fifty–two glaciers including five outlet glaciers of Jostedalsbreen. The record indicates minimum glacial input between ~6.7–5.7 ka, likely indicating that most GICs in the catchment had melted away (Vasskog et al., 2012; **Fig. 7**, record 5). The first reappearance of glaciers in the catchment may have occurred as early as ~5.7 ka. However, a

more well–defined Neoglacial is reported at ~4.2 ka, after which several intervals of significantly reduced glacial extent are reported (see Figure 8 in Vasskog et al., 2012). Nedre Sygneskardvatnet lake receives meltwater from Sygneskardbreen, a minor outlet glacier of NW Jostedalsbreen ice cap (Nesje et al., 2000; **Fig. 7**, record 6). The record shows that Sygneskardbreen existed in the catchment from deglaciation, ~10.3 ka, until ~7.3 ka. The glacier melted

away between ~7.3–6.15 ka, and subsequently has existed continuously (Nesje et al., 2000). Further east, a record from proglacial lake Vanndalsvatnet, indicates that Spørteggbreen glacier was absent from the catchment between ~8.6–2 ka, except for glacier episodes between roughly ~8.55–8.2, at ~7.9, 7.3, 7.15, and between 4.9–4.8 ka. Between ~2–1.4 ka, glacial episodes occurred at ~2, 1.9, 1.8, 1.7, 1.6, and 1.5 ka. Spørteggbreen has existed continuously since 1.4 ka (Nesje et al., 2006; **Fig. 7**, record 14). Shakesby et al. (2007) present a record from Liavatnet lake and stream–

bank mires that suggests that glaciers contracted prior to ~9 and until 8.4 ka (**Fig. 7**, record 4). A minerogenic layer suggests glacier expansion between ~8.4–7.8 ka, after which glaciers rapidly contracted. Glaciers were small between ~7.9–2.2 ka, except for short–lived glacier activity at ~5.6 and between ~3.7–3 ka. After ~2.2 ka, several possible glacier expansion events are noted, as well as a period of glacier contraction from ~1–0.7 ka (Shakesby et al., 2007). Evidence from Gjuvvatnet lake suggests that glaciers were present before ~10 ka, and absent from the catchment

between ~10 and 3.1 ka, except for between ~7.4–6.5 ka when glaciers were present (Karlén and Matthews, 1992; Matthews and Karlén, 1992; **Fig. 7**, record 8). Other glacial phases occurred between ~3.1–2.8, 2.7–2.6, 1.8–1.6 ka, and after ~0.7 ka. After roughly ~3 ka, the glaciers probably never disappeared entirely, except for possibly between ~1.3–0.9 ka, when organic content approached early–to–mid Holocene values (Karlén and Matthews, 1992; Matthews and Karlén, 1992). Next, a Holocene record of Leirbreen glacier from Bøvertunsvatnet lake suggests that the glacier

was small between ~10–8.4 ka (Matthews et al., 2000; **Fig. 7**, record 2). A glacier expansion episode is reported between ~8.4–8 ka, following which the glacier was absent from ~7.9–5.3 ka. Between ~5.3–2.5 ka, the glacier was either small or absent. Glacier growth began ~2.5 ka and the glacier varied in size between ~2.5–1.5 ka. Intensification of glacier growth occurred ~1.4 ka, however there was a short period of reduced glacier size between ~0.8–0.6 ka (Matthews et al., 2000). From the same study, a record of Liabreen and Høybreane glaciers from Dalsvatnet lake



suggests that glaciers were small before ~8.5 ka (Matthews et al., 2000; **Fig. 7**, record 3). Glaciers expanded between
~8.5–8.1 ka and were afterward absent between ~8 and 3.8 ka. Possible glacier variations and the first sign of
Neoglaciation occurred between ~3.8–1.4 ka, however, the period between ~2.2–1.8 ka was more certainly attributed
to a moderate glacier expansion. Intensification of glacier growth was reported after ~1.4 ka, however glaciers
contracted between ~0.8–0.4 ka (Matthews et al., 2000). Lie et al. (2004) report a record from lake

Bukkehåmmårtjørna of the Holocene fluctuations of Bukkehåmmårbreen glacier (**Fig. 7**, record 1). No glacier was in
the catchment between ~10.2–7.5 ka, after which there was a small glacier in the catchment until ~6.7 ka. Between
~6.7–6 ka the catchment was deglaciated, and after ~6 ka the glacier reformed and increased in size towards ~3.8 ka,
when it reached a size similar to present (Lie et al., 2004).

Finally, three additional records are available east and southeast of Bergen, Norway. Physical sediment variability
from a set of glacier–fed lakes shows that the ice cap of northern Folgefonna was present between ~11–9.6 ka (Bakke
et al., 2005b; Bakke et al., 2005c; **Fig. 7**, record 11). Between ~9.6 and 5.2 ka the ELA at northern Folgefonna was
above the highest mountain and no glacier was present in the catchment. Around 5.2 ka, the ice cap reformed, and
from ~4.6–2.3 the ice cap gradually grew toward its present extent and has existed in the catchment until present

(Bakke et al., 2005b; Bakke et al., 2005c). A record from lake Isdalsvatn indicates that the southwestern margin of the
plateau glacier, Hardangerjøkulen, was present until ~8.6 ka (Nesje et al., 1994; **Fig. 7**, record 9). The glacier was
absent from the catchment between ~8.6–3.8 ka, except for a readvance between ~7.8–7.6 ka. Glacier activity
commenced in the catchment again ~3.8 ka, but the glacier was small for some time. Increased glacial activity is
reported ~2.3 ka (Nesje et al., 1994). Inferences from two lacustrine records and terrestrial deposits indicate that N

Hardangerjøkulen glacier readvanced and was at a considerable size between ~8.5–8.3 ka **(**Dahl and Nesje, 1994; Dahl
and Nesje, 1996; **Fig. 7**; record 7). Subsequently, the glacier was absent between ~8.3–5.6 ka, except for a brief ELA
lowering ~6.2 ka. The glacier was generally small from ~5.6–1.2 ka, and continuous glacial input occurred from ~4.2
ka (Dahl and Nesje, 1994; Dahl and Nesje, 1996).

In summary, GICs first became smaller or absent altogether in Scandinavia in the early–to–middle Holocene, between
~10.2–7 ka. Combined, of the lake–based records available from Scandinavia, at least 80% indicate GICs were smaller
than present or absent by ~9.5 to 4.9 ka, except for a period of GIC readvance between ~8.5–7.9 ka, and near 100%
of GICs were smaller or absent between ~6.5–6.1 ka. There are no clear clusters in the timing of GIC regrowth in the
middle–to–late Holocene, however, the percent of GICs smaller or absent starts to decline roughly after ~6 ka, and

especially after ~4 ka. A review of Holocene glacier fluctuations in Scandinavia suggests that both the Scandinavian
ice sheet and local glaciers rapidly retreated during the early Holocene, but that the retreat was interrupted by periods
of GIC readvance in response to abrupt climate variations (Nesje, 2009). The review found that the period with the
most contracted glaciers occurred between ~6.6 and 6.3 ka, and that after ~6 ka, glaciers started to advance, with the
most extensive glaciers during the Little Ice Age (Nesje, 2009). Several potential periods of glacier advance are also

noted at ~8.5–7.9, 7.4–7.2, 6.3–6.1, 5.9–5.8, 5.6–5.3, 5.1–4.8, 4.6–4.2, 3.4–3.2, 3–2.8, 2.7–2, 1.9–1.6, 1.2–1, and 0.7–
0.2 ka (Nesje, 2009). Similarly, a review focusing on the Holocene history and future response of Norwegian GICs,



finds that most glaciers in Norway were completely melted away at least once due to high summer temperatures and/or reduced winter precipitation between ~8 and 4 ka (Nesje, 2008).



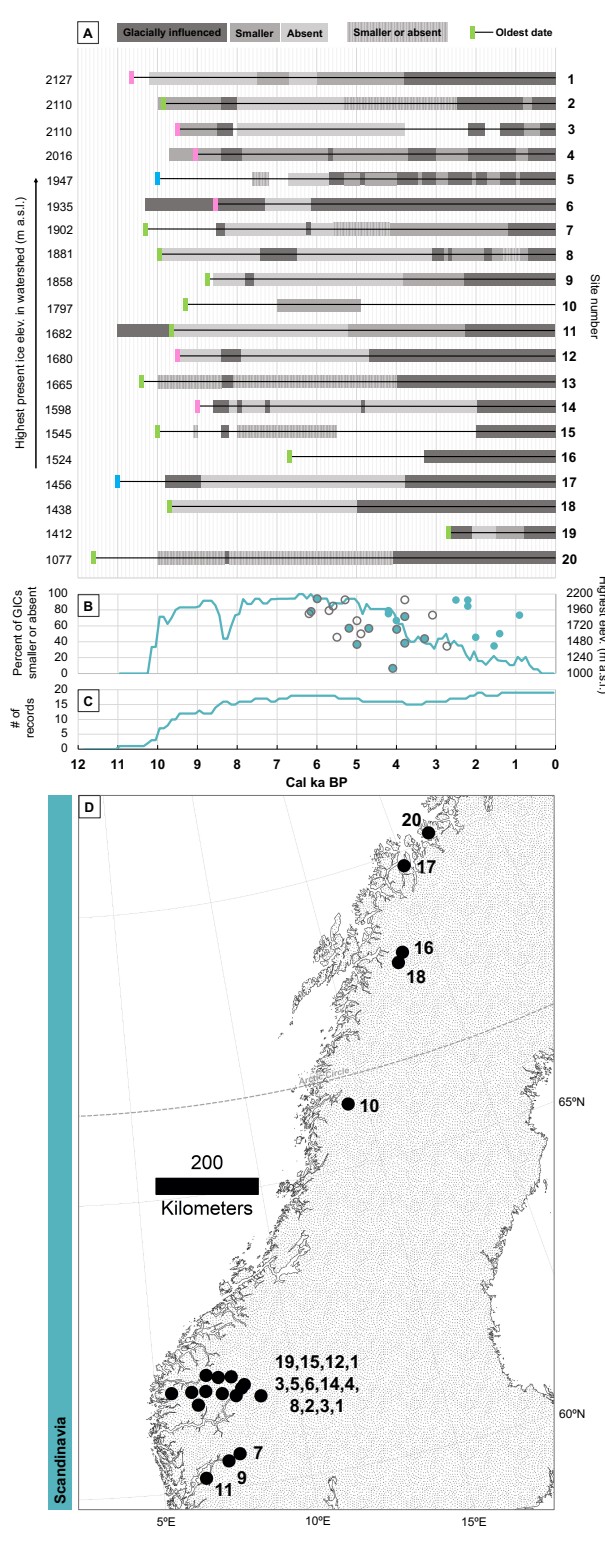



**Figure 7: Holocene lake–based GIC records from Scandinavia.** Panel descriptions, symbols and colors are the same as in Figure 3. Sites are: **1.** Lie et al., 2004 (lake Bukkehåmmårtjørna/Bukkehåmmårbreen glacier); **2.** Matthews et al., 2000 (Bøvertunsvatnet lake/Leirbreen glacier); **3.** Matthews et al., 2000 (Dalsvatnet lake/Liabreen and Høybreane glaciers); **4.** Shakesby et al., 2007 (Liavatnet lake/Sekkebreen, Tverrbyttnbreen, Kupenbreen, Kupbreen, Tverreggibreen, Greinbreen, Tverrbotnbreeen, Harbardsbreeen, and Fortundalsbreen glaciers); **5.** Vasskog et al., 2012 (lake Nerfloen/numerous glaciers (presently 52) including

five outlet glaciers of Jostedalsbreen glacier); **6.** Nesje et al., 2000 (Nedre Sygneskardvatnet lake/Sygneskardbreen glacier); **7.** Dahl and Nesje, 1994 & Dahl and Nesje, 1996 (lakes Hestebotnvatn and Tungevatn/N Hardangerjøkulen glacier); **8.** Karlén and Matthews, 1992 & Matthews and Karlén, 1992 (Gjuvvatnet lake/unnamed glacier); **9.** Nesje et al., 1994 (lake Isdalsvatn/SW Hardangerjøkulen glacier); **10.** Bakke et al., 2010 (Vestre– and Austre Kjennsvatnet/Austre Okstindbreen glacier); **11.** Bakke et al., 2005b & Bakke et al., 2005c (Dravladalsvatn, Vassdalsvatn, Vetlavatn, and Vassdalsvatn lakes/N Folgefonna); **12.** Seierstad et

al., 2002 (lake Grøningstølsvatnet/Grovabreen glacier); **13.** Nesje et al., 2001 (lake Jarbuvatnet/Flatebreen glacier); **14.** Nesje et al., 2006 (lake Vanndalsvatnet/Spørteggbreen glacier); **15.** Bakke et al., 2013 (lake Nedre Sørsendalsvatn/Blåbreen glacier); **16.** Snowball and Sandgren, 1996 (lakes Bajimus Gorsajávri, Vuolimus Gorsajávri, and Gaskkamus Gorsajávri/Kårsa glacier); **17.** Bakke et al., 2005a (lake Aspvatnet/Lenangsbreene glacier); **18.** Rosqvist et al., 2004 (Vuolep Allakasjaure lake/unnamed glacier); **19.** Nesje et al., 1995 (lake Grøndalsvatn/Ålfotbreen glacier); **20.** Wittmeier et al., 2015 (lake Jøkelvatnet, Store Rundvatnet, and

Storvatnet/Langfjordjøkelen ice cap). Glacier elevations for Shakesby et al. (2007) were estimated via the ASTER GDEM due to data holes in the ArcticDEM.

### 3.6 Svalbard

The Svalbard archipelago lies in the polar North Atlantic and extends from ~74–81°N. Roughly 60% of the land is covered with GICs. The climate of Svalbard is characterized as dry high Arctic, but is generally milder, wetter, and cloudier than other areas at comparable latitudes (Hanssen–Bauer et al., 2019). Svalbard's relatively mild climate is largely due to the transport of heat associated with the warm West Spitsbergen Current (WSC), which flows northward along Svalbard's western coast (Hanssen–Bauer et al., 2019). The release of heat from the WSC is especially

influential on Svalbard's climate during winter and affects sea–ice concentration (Hanssen–Bauer et al., 2019). Generally, the archipelago's coastal northeast is colder than the areas to the south and southwest (Hanssen–Bauer et al., 2019). Summer (June, July, and August) average temperature ranges from ~3.4 to 4.2°C (Førland et al., 2011, weather station data collected between 1961–1990), and average estimated annual precipitation is ~720 mm (Hanssen–Bauer et al., 2019).


The six lake–based records of Holocene GIC fluctuations available from Svalbard are located on the west and north coast of Spitsbergen, the archipelago's largest island (**Fig. 8**). The two lake's which host the highest elevation GICs within their watersheds today (i.e., Allaart et al., 2021; Røth et al., 2018, **Fig. 8**, records 1 & 2) show similar glacial histories through the Holocene. The northern most record (i.e., Allaart et al., 2021) is from Femmilsjøen, one of

Svalbard's largest lakes, which receives runoff from Longstaffbreen, an outlet of Åsgardfonna ice cap. The record indicates that glaciers existed in Femmilsjøen's catchment after the lake's isolation from the sea (between ~11.7–11.3 ka) until ~10.1 ka. Between ~10.1–3.2 ka, glacial meltwater input ceased, and the ice cap is interpreted to have been greatly reduced or entirely disappeared. At ~3.2 ka, glacial input commenced, indicating glacier regrowth in the lake's



catchment. The ice cap is interpreted to have reached a size no smaller than the present extent by ~2.1 ka (Allaart et
al., 2021). Roughly 28 km to the SW, and across Wijdefjorden, the record from lake Vårfluesjøen, which currently
hosts four glaciers in its catchment (the largest, Uglebreen), suggests that glaciers were small or absent from the onset
of lacustrine sedimentation at ~10.2 ka until 4.2 ka (Røth et al., 2018). Glacier activity in the catchment is reported
again at ~3.5 ka (Røth et al., 2018). On the southern coast of Isfjorden, in western Spitsbergen, a Holocene record
(**Fig. 8**, record 3) from the east basin of lake Linnèvatnet, suggests that Linnèbreen glacier was absent between ~11.25–
4.95 ka (Svendsen and Mangerud, 1997). The glacier is interpreted to have started to reform as early as ~4.95 ka and
is reported to have existed continuously from ~3.45 ka, the time of the first appearance of a glacial lamination in the
lake record (Svendsen and Mangerud, 1997). On Mitrahalvøya peninsula, Røth et al. (2015) present a Holocene record
of glacier activity from Kløsa lake (**Fig. 8**, record 4). The record indicates that Karlbreen glacier was small or had
completely melted away between ~9.2–3.5 ka and was smaller than present between ~1.4–1.2 ka. The Neoglacial is
interpreted to have begun ~3.5 ka, when the reconstructed glacier ELA shows a significant lowering (Røth et al.,
2015). On the northern–west coast of Spitsbergen, a sediment record from lake Gjøavatnet shows that Annabreen
glacier likely persisted in the lake's catchment until ~8.4 ka (de Wet et al., 2018; **Fig. 8**, record 5). Between ~8.4–1
ka, organic–rich sediment was deposited, indicating the glacier was smaller or absent. Minerogenic input to lake
Gjøavatnet from Annabreen glacier abruptly began ~1 ka and has continued to the present (de Wet et al., 2018).
Located very close by to Kløsa lake (i.e., Røth et al., 2015, record 4), lake Hajeren is currently fed by two northwest–
facing cirque glaciers (van der Bilt et al., 2015; **Fig. 8**, record 6). The lake record suggests that glaciers were present
in the catchment following deglaciation (prior to ~11.3 ka) and until ~7.4 ka. Glaciers were absent between ~6.7–0.7
ka except for three centennial scale glacier advances at ~4.25–4.05 (marking the onset of the Neoglacial), ~3.38–3.23,
and ~1.1–1.0 ka. Sustained glacier presence is reported from ~0.7 ka. Finally, a second record from lake Linnèvatnet
records Holocene glacier activity in a western cirque which is not glaciated at present and contains only remnant
stagnant ice (Snyder et al., 2000). The record indicates that the cirque was deglaciated prior to the lake's isolation,
~10.3 ka, and that the cirque remained free of ice until the Little Ice Age, or sometime between 0.6–0.4 ka (although
the chronology is uncertain) (Snyder et al., 2000). We exclude this record from our compilation due to the uncertain
late Holocene chronology and because the studied cirque which feeds the west basin of lake Linnèvatnet is not
currently glaciated, and thus, ice absence cannot be used to infer warmer than present conditions.

In summary, GICs first became smaller or absent altogether in Svalbard in the early–to–middle Holocene, between
~11.25–7.4 ka. Of the GIC records available, at least 80% indicate that GICs were smaller or absent between ~8.4–
3.5 ka, and at least 60% indicate that GICs were smaller or absent from ~10.1 ka. In the middle–to–late Holocene,
GICs first regrew in lake catchments as early as ~4.95 ka (i.e., Svendsen and Mangerud, 1997). Sustained regrowth
of GICs occurred between ~3.5–0.7 ka. Two main clusters GIC regrowth are evident in the Svalbard lake records (**Fig.
8B**). Lake catchments that host high elevation GICs (between ~635–883 m a.s.l.) show sustained regrowth of ice ~3.5
ka BP, while lake catchments that host lower elevation GICs (~475 m a.s.l. and below) show sustained regrowth of
ice after ~1 ka. These inferences agree well with a recent synthesis of glacier activity on Svalbard which infers that





710    the Holocene glacial minimum occurred sometime between ~8 and 6 ka, and that glacier re–advances predominantly
occurred between ~4–0.5 ka, with the highest frequency between ~1–0.5 ka (Farnsworth et al., 2020).

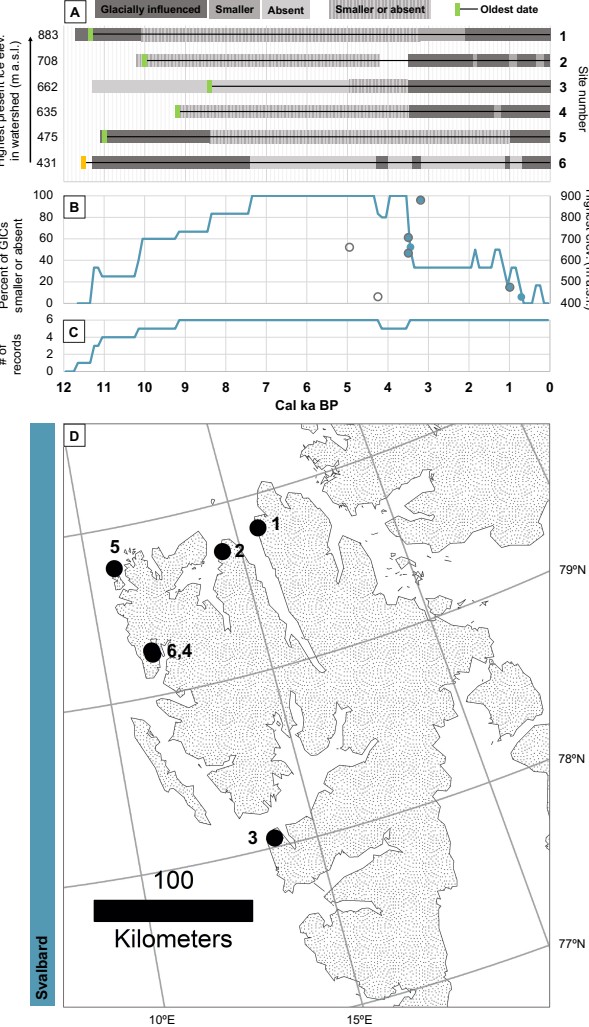

**Figure 8: Holocene lake–based GIC records from Svalbard.** Panel descriptions, symbols and colors are the same as in Figure
715    3. Sites are: **1.** Allaart et al., 2021 (lake Femmilsjøen/NW Åsgardfonna Ice Cap); **2.** Røth et al., 2018 (lake Vårfluesjøen/Bybreen,
Uglebreen, Salomonbreen, and Grennabreen glaciers); **3.** Svendsen and Mangerud, 1997 (lake Linnèvatnet, East basin/Linnèbreen
glacier); **4.** Røth et al., 2015 (Kløsa lake/Karlbreen glacier); **5.** de Wet et al., 2018 (lake Gjøavatnet/Annabreen glacier); **6.** van der
Bilt et al., 2015 (lake Hajeren/North and South glaciers). The oldest ages reported in Aallart et al. (2021) are marine. We show the
oldest age reported in the lacustrine section of the record.

720

**3.7 Russian Arctic**



Stretching over roughly 24,000 km of coastline, the Russian Arctic is vast and includes numerous archipelagos which are dispersed throughout the marginal seas of the Arctic Ocean. The most prominent archipelagos include: The 192 islands of Franz Josef Land, located ~80°N; Novaya Zemlya, an extension of the northern Ural Mountains, consisting of two major islands between the Kara and Barents Sea; and Severnaya Zemlya, which is comprised of four major islands and lies in the Laptev Sea, off Siberia's Taymyr Peninsula. Around the Polar Ural Mountains, the climate is cold and continental with mean summer temperatures ~7°C and annual precipitation ~600 mm (Svendsen et al., 2019; Solomina et al. 2010). Franz Josef Land is ~85% glaciated and has a high Arctic climate with mean annual air temperatures around –13°C (Lubinsky et al., 1999). Severnaya Zemlya has a very harsh cold and dry polar desert climate, with mean monthly July and August temperatures of near ~0°C (Andreev et al., 2008).

The Holocene history of GICs in the Russian Arctic is very poorly known, and only two lake–based records of GIC fluctuations are available. Thus, we also include summarized results from Lubinsky et al. (1999), who provide the most comprehensive overview of Holocene GIC fluctuations via 45 radiocarbon ages from 16 glacier margins in Franz Josef Land (**Fig. 9**, record 3). The ${}^{14}$C ages from driftwood, whalebones, shells, and mosses indicate that many glaciers were behind their present margins before ~10.7 ka (and as early as ~12 ka) and remained so until at least ~5 ka (Lubinsky et al., 1999; **Fig. 9**, record 3). Subsequently, glaciers expanded, probably reaching their present margins by at least ~3.5 ka, and certainly by ~2.1 ka (Lubinsky et al., 1999). We note that in the original publication the uncalibrated laboratory ages of the marine samples had been corrected by subtracting 440 years from the reported age (Lubinsky et al., 1999). We add 440 years to get the original laboratory ages and then calculate the ΔR value and uncertainty with the Marine20 database using the nearest 10 data points to Franz Josef Land before calibration.

On October Revolution Island, sediment and peat profiles from Changeable lake and the Ozernaya River basins indicate that in the early Holocene the climate was warmer than today, and that Vavilov Ice Cap was at or behind its present margins between ~11.5–9.5 ka (Andreev et al., 2008; **Fig. 9**, record 2). Sediment records from lake Bolshoye Shchuchye in the Polar Ural Mountains suggest that GICs had melted away completely by ~15–14 ka (Svendsen et al., 2019; Haflidason et al., 2019; **Fig. 9**, record 1). Between ~4–3 ka, the influx of sediments into the lake basin increased, suggesting glacier growth in the catchment area of lake Bolshoye Shchuchye (Haflidason et al., 2019).

In summary, the limited number of records of GIC variability in the Russian Arctic indicate that GICs first became smaller than present or absent from at least the very early Holocene, and probably before, between ~14.5–11.5 ka. GICs remained small or absent through most of the middle Holocene and began to regrow between roughly ~5 and 4 ka.

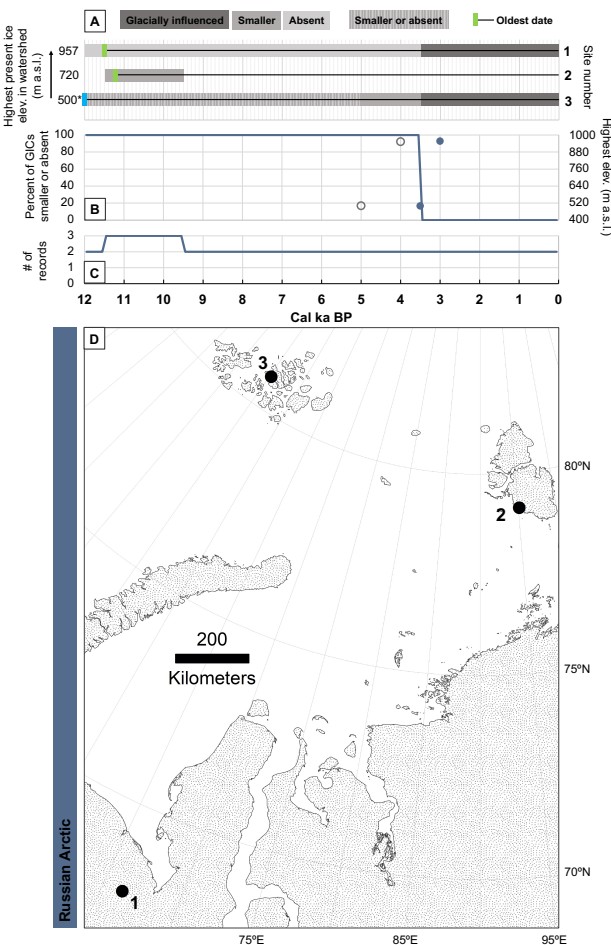

**Figure 9: Holocene lake–based GIC records from the Russian Arctic.** Panel descriptions, symbols and colors are the same as in Figure 3. Sites are: **1.** Svendsen et al., 2019 & Haflidason et al., 2019 (lake Bolshoye Shchuchye/unnamed glaciers); **2.** Andreev et al., 2008 (Changeable lake/Vaviloc ice cap); **3.** Lubinski et al., 1999 (non–lake record/review of 16 glacier margins in Franz Josef Land). We note that the oldest reported age from lake Bolshoye Shchuchye (i.e., Svendsen et al., 2019 & Haflidason et al., 2019) is pre–Holocene. We report the oldest Holocene age. At present, there are a few very tiny cirque glaciers around the lake, however the theoretical snowline in the Polar Urals is already well above the highest mountain peaks (Svendsen et al., 2019). In addition, the timing of glacier regrowth is reported sometime between ~4–3 ka. We use an average for the timing of regrowth (i.e., 3.5 ka) in panel A, but use 4 ka as the earliest regrowth and 3 ka as the timing of sustained regrowth in panel B. *We report the average present summit elevation of ice caps in Franz Josef Land for the non–lake record summary in panel A.

### 3.8 Pan–Arctic

Overall, our compilation of Holocene lake–based GIC records spanning the Arctic (**Fig. 10D**) shows that the majority (50% or more) of the studied GICs were smaller than present or absent between 12–11.1 ka and 10–3.4 ka, and that





most (80% or more) were smaller than present or absent between 7.9–4.5 ka (and before ~11.7 ka BP, however only two records extend back 12 ka and both are from the Russian Arctic). The pan–Arctic dataset also indicates a relatively abrupt increase in the percent of GICs smaller than present or absent at ~10 ka. The percent of GICs smaller or absent

peaks in the middle Holocene between ~7.1–5.7 ka, when greater than 90% of GICs were smaller or absent. In the middle–to–late Holocene (after ~6 ka), GICs began to regrow across the Arctic, although the timing of individual GIC regrowth is variable (**Fig. 10D**). We find a weak relationship between the timing of earliest GIC regrowth and GIC elevation (GICs with the highest present–day ice elevation in their catchments tended to regrow earliest) (**Fig. 10D**). The combined records also indicate two periods of intensified GIC growth in the late Holocene (denoted by more

abrupt drops in the percent of GICs smaller than present or absent) between ~4.5–3 ka and after 2 ka (**Fig. 10D and E**).

We estimate an approximate average magnitude of Arctic–wide summer warming of at least ~2±1.3°C above present using the difference in elevation between the GICs' present steady–state ELAs (calculated here using an AAR ratio

of 0.67) and their highest present ice elevations, multiplied by a standard lapse rate of 6.5°C/km (after Larocca et al. 2020a; 2020b). This calculation excludes the GICs that were reported to have survived early–to–middle Holocene warmth and includes all GICs that became smaller than present or absent at some point in the early or middle Holocene. Since this approximation is the average summer temperature rise needed to elevate ELAs above modern ice surfaces but ELAs may have risen higher than that minimum threshold, and these calculations assume no change in

precipitation (whereas the HTM is generally believed to have been wetter on average, e.g., Thomas et al., 2016; 2018), it represents a minimum constraint on average Arctic warming during the early Holocene (roughly between 10–8 ka), when most Arctic GICs first became smaller or absent.

Our Arctic dataset agrees with a global review of Holocene glacier fluctuations which found that Northern

Hemisphere, mid–to–high latitude glaciers were smaller than present or at least equal to their present sizes ~8–4 ka (Solomina et al., 2015). Similarly, a recent study investigating Neoglacial cooling in the Arctic developed a simple index to summarize the relative extent of GICs in Arctic regions (the status of which were derived from the above global review) and reported that GICs were retreated throughout the Arctic ~8–6.5 ka (McKay et al., 2018). The study suggested that this period of uniform retreat, and by inference widespread warmth, can be attributed to high summer

insolation combined with the absence of (or lessening influence of) the Laurentide Ice Sheet (LIS) (McKay et al., 2018), which lingered well after peak insolation, as late as ~7 ka.

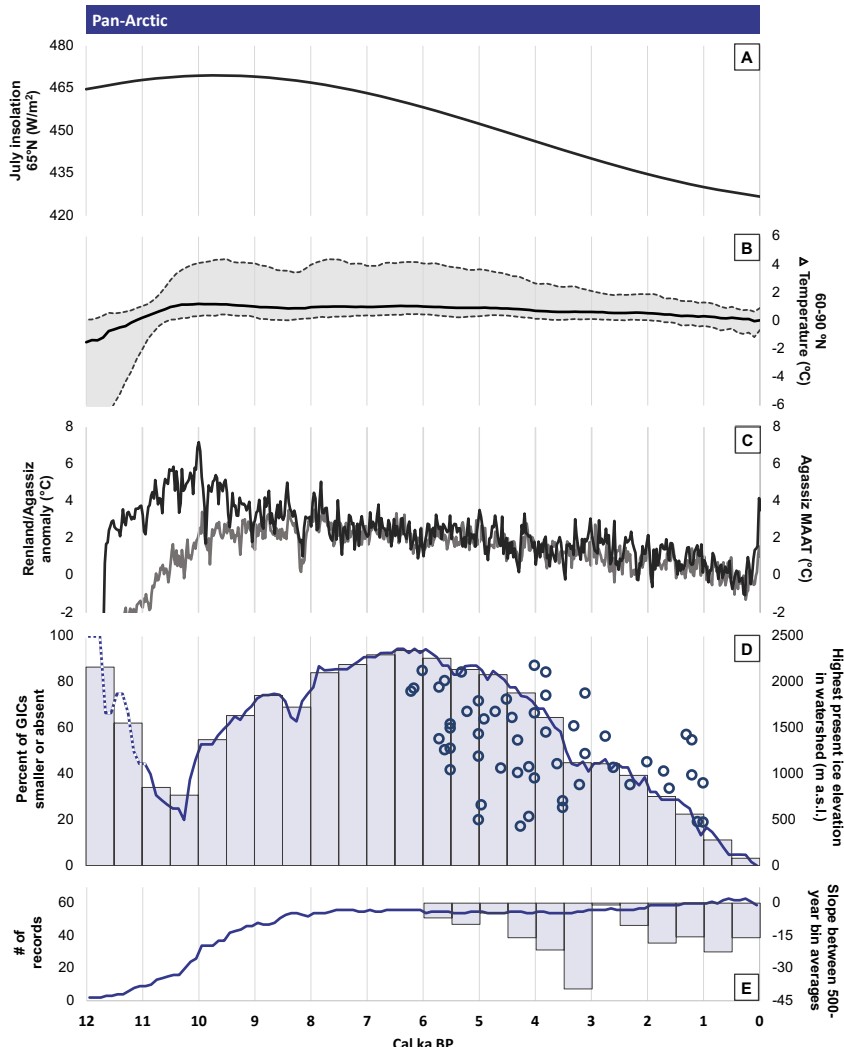

**Figure 10: Pan–Arctic summary of lake–based GIC records. A.** July insolation at 65°N. **B.** Reconstructed mean annual temperature from the Temperature 12k database for the latitudinal band 60–90°N, the multi–method ensemble median (solid line) and 5th and 95th percentiles (lower and upper dashed lines). Temperature anomalies are relative to 1800–1900 CE (Kaufman et al., 2020). **C.** Gray line is a Holocene temperature reconstruction based on elevation corrected average $\delta^{18}O$ data from Renland & Agassiz ice cores. Temperatures are 20–year averages given as deviations from a smooth estimate of present temperatures in Greenland (Vinther et al., 2009). Black line is the elevation–corrected Agassiz $\delta^{18}O$ mean annual air temperature reconstruction (Lecavalier et al., 2017). **D.** Line shows the percent of GICs smaller than present or absent from 12 to 0 ka, calculated in 100–year bins. Dashed line show time bins in which a low number (<10) of records were available. Bars show 500–year bin averages. Open dots show the timing of earliest GIC regrowth in the middle–to–late Holocene versus the highest present ice elevation inside the studied lake's watershed. **E.** Line shows the number of records from 12 to 0 ka. Bars show the slope between the 500–year bin





averages (above) from the middle–to–late Holocene. More negative slopes indicate time periods with a higher number of GICs regrowing.


## 4 Discussion

### 4.1 The early to middle Holocene: when were summers warmer in the Arctic?

Early to middle Holocene climate in the northern high latitudes was modulated by the interplay between climate forcings, changing surface boundary conditions, and resulting feedbacks. Summer insolation peaked in the early Holocene and was up to ~10% higher than today at 60°N in June, while decaying Northern Hemisphere ice sheets altered local albedo and regional atmospheric circulation, and discharged cold freshwater and icebergs into adjacent seas, modifying sea surface conditions and ocean circulation (Kaufman et al., 2004; Jennings et al., 2015). Broadly 825   speaking, summer temperatures in the Arctic appear to have peaked ~1–3°C above 20[th] century averages in the first half of the Holocene and were largely driven by orbital forcing, however, the timing and magnitude of peak warmth varied geographically across the Arctic, differing by thousands of years and by several degrees between locations (Kaufman et al., 2004, 2016; Kaplan and Wolfe, 2006; Miller et al., 2010; Briner et al., 2016; Sejrup et al., 2016; Axford et al., 2021). This marked variability speaks to the complexity of the climate system's response to insolation, 830   other forcings, and feedback mechanisms.

Our compilation of lake–based records shows strong coherence across the Arctic at a broad scale and suggests that summer air temperatures were likely warmer than present (as inferred from GICs smaller than present or absent) across the majority of the Arctic in the early Holocene, from at least ~10 ka, and that most places were warmer than present 835   by at least ~8 ka. The regional GIC evidence suggests even earlier summer warmth in some areas, especially in the Russian Arctic and in Svalbard, where GICs first became smaller or absent between ~14.5–11.5 ka and ~11.25–7.4 ka respectively. Although we find evidence for the onset of warmer–than–present summers across most of the Arctic in the early Holocene, our compilation shows the highest percentage of GICs smaller than present or absent during the middle Holocene (>90% smaller than present or absent) from ~7–6 ka (**Fig. 10D**). We suggest that this pattern 840   reflects the asynchronous timing and magnitude of peak Holocene warmth across the Arctic (e.g., Kaufman et al., 2004; Briner et al., 2016) and does not necessarily indicate that the warmest part of the Holocene occurred in the middle Holocene uniformly. Instead, these data more likely indicate that the mid–Holocene was a more spatially consistent period of warmer than present summers across the Arctic as a whole. The complex boundary conditions of the early Holocene, including residual Pleistocene ice sheets, likely modulated regional and sub–regional climatic 845   response to changing insolation and greenhouses gases. It is also possible that counteracting effects on glacier mass balance from increased precipitation were stronger in the early Holocene in some locations. Given the pronounced asymmetries in early Holocene climate, somewhat expectedly we find substantial variability in the timing that GICs first became smaller or absent between sites within our defined geographic regions (**Fig. 11**). This variability may be due to significant climatic differences across the (sometimes geographically large) regions causing differing responses 850   to temperature and/or precipitation change, differing sub–regional and local–scale responses to early Holocene



forcings and feedbacks and to changing atmosphere–ocean–sea ice dynamics, and to differing glacier or site–specific characteristics.

The onset of summer warmth in the early Holocene in most of the Arctic is supported by a wide variety of evidence
from terrestrial and marine archives (e.g., Kaufman et al., 2004; Miller et al., 2010). In addition, a recent review of Holocene temperature reconstructions from around Greenland found that when accounting for some key issues with proxy interpretation, much of Greenland (with the possible exception of south Greenland) experienced warmer than present summers in the early Holocene by ~10 ka (Axford et al., 2021). Although reflective of annual temperature, elevation–corrected reconstructions from ice cores based upon stable isotopes of ice suggest early peak warmth. The
elevation–corrected temperature reconstruction derived from average $\delta^{18}O$ data from the Renland and Agassiz ice cores (located in central–east Greenland and on Ellesmere Island in Canada, respectively) shows peak warmth in the early Holocene, from ~9.5 to 7.5 ka with annual temperature deviations up to ~3.5°C above the smoothed estimate of present temperatures in Greenland (**Fig. 10C**; Vinther et al., 2009). A revised temperature reconstruction from Agassiz Ice Cap suggests even earlier and stronger peak warmth from ~11 to 8 ka, peaking at 6.1°C at ~10 ka (with an applied
Gaussian low pass filter σ =50 yrs; 2σ uncertainty 4.3–8.3°C) (**Fig. 10C;** Lecavalier et al., 2017).  Finally, using an extensive multi–proxy database of paleo–temperature time series, Kaufman et al. (2020) reconstruct mean annual surface temperature over the Holocene for the globe and for six 30° latitudinal bands using five different statistical methods. The multi–method ensemble median for the latitudinal–zone 60–90°N, shows the warmest interval ~10.1–9.7 ka with annual mean surface temperatures ~1.2°C (0.4, 4.2) warmer than the 19th century (5th, 95th percentiles)
(Kaufman et al., 2020; **Fig. 10B**).

Both proxy–based and climate model–based studies have concluded that the disintegrating LIS counteracted early Holocene insolation–driven warming and caused a subdued and/or delayed warming over the northwest North Atlantic, particularly in the Baffin Bay–Labrador Sea region, while summer temperatures in other regions followed
orbital forcing (Mitchell et al., 1988; Kaufman et a., 2004; Renssen et al., 2009; Briner et al., 2016). Long–term suppression of summer temperatures around Baffin Bay through the early Holocene is not clear in our synthesis (though there are very few lake–based records of GIC change there). However, the influence of early Holocene abrupt cold event(s) attributed to meltwater pulses from the collapsing LIS are apparent in our compilations between 9 and 8 ka (**Fig. 10D**). Some GICs reappeared in their catchments during the 8.2 ka cooling event (associated with the
catastrophic drainage of ice dammed glacial lakes in the Hudson Bay area and collapse of the Hudson Bay Ice Saddle, e.g., Hoffman et al., 2012; Lochte et al., 2019). Interestingly, some studies record GIC advance prior to ~8.2 ka BP, supporting the idea of a broader (~160–400 year) event as seen in other proxy records (Lochte et al., 2019). It is also possible that radiocarbon dating uncertainties contribute to this spread.

**4.2 The middle to late Holocene: when did summer cooling commence in the Arctic?**



In the northern high latitudes, summer cooling from the middle to late Holocene was primarily driven by the slow and steady decline in summer insolation, partially offset by radiative forcing by greenhouse gases which rose through the middle to late Holocene (Ramaswamy et al., 2001; McKay et al., 2018). Recent Arctic–focused paleoclimate syntheses

and modeling studies have highlighted regional asymmetries in cooling onset, as well as differences in the rate and magnitude of cooling between regions, and between larger–scale areas of the Arctic (e.g., between the Pacific and Atlantic sectors) (McKay et al., 2018; Zhong et al., 2018). Thus, other forcings and feedbacks, such as volcanic eruptions, solar activity, sea ice expansion, changes in terrestrial snow cover and in ocean circulation, probably also played a major role in the expression of the Neoglacial regionally, and/or caused sub–millennial–scale climate

variations superimposed on the progressive orbitally driven cooling trend in the second half of the Holocene (e.g., Miller et al., 2012; Solomina et al., 2015; McKay et al., 2018; Zhong et al., 2018). Yet, much about the mechanisms for, and relationship between these climate drivers, feedbacks, and GIC fluctuations remains uncertain.

Across the Arctic, GICs in our compilation first began to regrow or expand in the middle Holocene, mostly after ~6

ka (**Fig. 10D**). By roughly ~3.5 ka, less than 50% of studied glaciers were smaller than present or absent, and by roughly ~1.2 ka less than 20% were smaller than present or absent (**Fig. 10D**). This suggests that summer air temperatures began to cool in some areas from at least ~6 ka, and that summer temperatures were cool enough to support erosive GICs in the majority of the Arctic by ~3.5 ka, and across most of the Arctic by ~1.2 ka. The regional lake–based GIC evidence suggests substantial variability in the timing of earliest GIC regrowth between and within

regions (**Fig. 11**). Our regional compilations show earliest GIC regrowth in Scandinavia and in the Russian Arctic, where GICs first regrew in lake catchments in the middle Holocene, as early as ~6.2 ka in Scandinavia (**Fig. 11**). Similarly, in a review of global Holocene and late Pleistocene alpine glacier fluctuations, Davis et al. (2009) find that glaciers reformed and/or advanced beginning as early as 6.5 ka in some areas, and consistent with our inferences, McKay et al. (2018) find an early onset of cooling in Fennoscandia and Russia.


Our results suggest a weak relationship between the timing of earliest GIC regrowth and the highest present ice elevation inside the studied lake's watershed (**Fig. 10D**). This reflects that topography had some influence on the pattern of GIC regrowth (probably within very small geographic areas, e.g., Larsen et al., 2017; Larocca et al., 2020b) and that very broadly as summer temperature declined, ELAs lowered to intersect the local landscape where GICs

regrew on the highest peaks first. Although differences in topography might help to explain some of the variability within regions, the weakness of this relationship on an Arctic–wide scale (and even within geographically large regions) supports the findings of McKay et al. (2018): namely, that the timing of GIC regrowth is not simply a local threshold effect, and is associated with other key controls, such as increased cooling rates, the timing of which differed between regions, probably at least in part due to regional climate dynamics (McKay et al., 2018). Two periods of

intensified glacier growth (indicated by the change in slope between 500–year bin intervals; **Fig. 10D and E**) are apparent in our pan–Arctic compilation, between ~4.5–3 ka and after ~2 ka. These periods align well with other evidence of Neoglaciation in the Arctic and potentially suggest periods with higher rates of cooling. Consistent with the GIC evidence presented here, McKay et al. (2018) find no evidence for a synchronous Arctic–wide onset of cooling





but find two major pulses of Neoglacial advance from the GIC evidence—the first between ~4.5–2 ka, and the second beginning at ~2 ka and culminating in the LIA (McKay et al., 2018). Furthermore, the timing of these two identified intervals of Neoglacial onset are consistent with times of accelerated Holocene cooling (McKay et al., 2018). In addition, radiocarbon ages of ice–entombed plants from Svalbard suggest snowline lowering from at least 4–3.4 ka, and progressive, but episodic, lowering since then, indicating a significant response to other forcings and/or substantial internal climate variability in addition to insolation forcing (Miller et al., 2017). Solomina et al. (2015) find numerous Neoglacial advances after roughly 4 ka, as well as clustering of glacier advances which correspond to cooling in the North Atlantic, at 4.4–4.2 ka, 3.8–3.4 ka, 3.3–2.8 ka, 2.6 ka, 2.3–2.1 ka, 1.5–1.4 ka, 1.2–1.0 ka and 0.7–0.5 ka, and roughly correspond to multidecadal periods of low solar activity or volcanic eruptions. Similarly, in comparing glacial records across the Northern Hemisphere, Bakke et al. (2010) find common glacier advances centered at roughly 4.0 ka, 2.7 ka, 2.0 ka, 1.3 ka and during the LIA.

In summary, it is becoming clear that the Arctic did not cool synchronously from the middle to late Holocene despite the smooth, near–linear decline in Northern Hemisphere summer insolation—the primary driver of climate change during this period. Instead, the current body of evidence suggests a stepwise–like cooling (i.e., with times of intensified and/or accelerated cooling) as well as significant variability in the onset and rate of cooling regionally and sub–regionally. This implies that other mechanisms, forcings, and feedbacks, other than orbitally driven insolation, were important contributors to the regional and Arctic–wide expression of Neoglacial cooling.

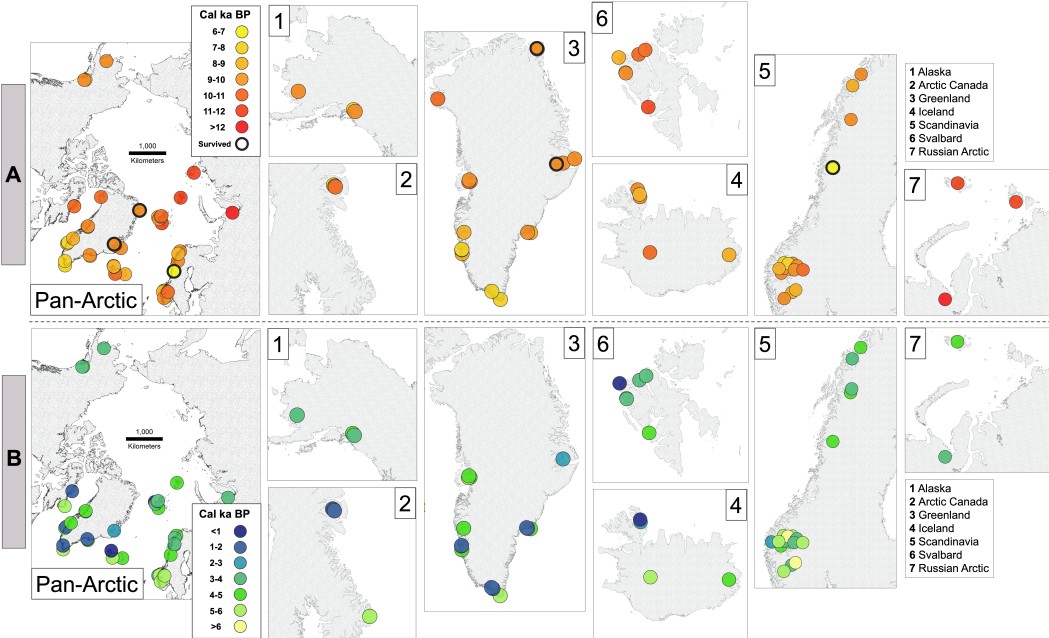

**Figure 11: Spatial summary of Holocene lake–based GIC records. A.** When each study first reported GICs smaller than present or absent, indicating a minimum bound on warmer than present conditions in the early–to–middle–Holocene. Black open circles



denote GICs that are reported to have likely survived the HTM. **B.** When each study reported the earliest instance of GIC presence in the lake catchment, indicating cooling and more specifically that local ELA first dipped below the highest local topography. We exclude 12 records from panel A and 13 records from panel B that do not report this information reliably (i.e., threshold–lake records that only record GICs larger than today, short records, and unclear and/or uncertain records). In addition, Thomas et al.
(2010) report that Longspur lake received glacial meltwater through the Holocene, implying that some glaciers survived, however the record is excluded here because if/when the glaciers were smaller than present is not reported.

**5 Conclusions**

Lake–based GIC records reveal that many Arctic GICs shrank smaller than today beginning in the early Holocene, suggesting warmer than present summer air temperatures around much of the Arctic quite early in the Holocene. This was followed by near–ubiquitous GIC retraction in the middle Holocene, between ~7–6 ka, when nearly all the records indicate smaller than present or absent GICs. This could but does not necessarily imply higher summer temperatures in the middle Holocene than in the early Holocene. Indeed, prior work using different proxies has found peak summer
temperatures in the early Holocene in some parts of the Arctic, but also a more spatially and temporally variable climate versus the middle Holocene (e.g., Kaufman et al. 2016; Lecavalier et al. 2017; Axford et al. 2021). We suggest that the ubiquity of glacier retraction across the Arctic in the middle Holocene reflects more spatially widespread, consistent summer warmth than in the early Holocene. GICs subsequently reformed and grew through the middle to late Holocene, with GIC regrowth beginning at some sites at or before ~6 ka. Two periods of intensified GIC growth
occurred between ~4.5–3 ka and after 2 ka, probably indicating intervals of intensified cooling. Although Arctic–wide patters emerge, overall, our results indicate pronounced spatial and temporal variability in the timing of early–to–middle Holocene warming, and middle–to–late Holocene cooling. Finally, our early Holocene estimate of Arctic–wide summer warming of at least ~2°C above present is consistent with previous syntheses of paleotemperature evidence from the Arctic. Our synthesis therefore reinforces that relatively modest summer warming (compared with
projections of larger future climate change, e.g., Collins et al., 2013) drove major environmental changes across the Arctic including the widespread loss of GICs. This knowledge foretells the continued rapid retreat and eventual disappearance of most of the Arctic's small GICs. Along with environmental impacts, this loss will have numerous sociocultural and economic ramifications for many Arctic communities in the coming decades (e.g., Huntington et al., 2019). More studies that model projected GIC loss at regional to sub–regional scales are needed to provide a more
comprehensive view of future Arctic change and to address these often overlooked local implications.

**Data availability:** Supplementary data, Holocene lake-based Arctic glacier and ice cap records, is archived at the NSF Arctic Data Center [DOI TBD after revision].

**Supplement link:** [TBD after revision].

**Author contribution:** LL and YA designed the study and prepared the manuscript. LL compiled the data.

**Competing interests:** The authors declare that they have no conflict of interest.



**Acknowledgments**

We first acknowledge Arctic Indigenous Peoples who have lived on and stewarded Arctic lands for thousands of years, and today, whose communities are disproportionately affected by climate change and glacier loss. This research was supported by the U.S. National Science Foundation's Office of Polar Programs (CAREER Award 1454734) and Geography and Spatial Sciences Program (DDRI Award 1812764). We thank Peter Puleo and Regan Steigleder who provided helpful comments and discussion.

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

1455    497.