# Peer review of "Arctic glaciers and ice caps through the Holocene: A circumpolar synthesis of lake-based reconstructions"

_Climate of the Past, 2021_

## Author Response (AR1)

We thank the two reviewers for detailed and constructive reviews that improved our manuscript. Our revisions are described point-by-point below.

**Reviewer #1**

The manuscript submitted by Larocca and Axford falls well within the scope of CP and presents a clear, well-organized and illustrated synthesis of a very specific proxy for Holocene climate change in the Arctic: lake-based reconstructions that document the growth and decay of adjacent glaciers and ice caps (GICs). In my opinion, the novelty and originality of this study is reflected in its consideration of only this particular proxy, from many different regions of the circumpolar Arctic. Although I found myself wishing for more background information and complementary studies that could add weight to some of the interpretations presented, and for the addition of records from regions where existing studies were rejected due to weak dating control, for example, I think that because this manuscript covers such a large region (the circumpolar Arctic), sticking to precisely defined criteria is critical. The conclusions of the study are timely and pack some punch, as they clearly indicate that the modest warming of the early Holocene (well below that which is forecasted for Arctic regions in the future) was enough to cause widespread partial or full retreat of GICs; with ongoing climate warming, amplified in Arctic regions, many to most Arctic GICs will clearly continue retreating until they disappear entirely. The conclusions of this study are also in general agreement with other studies of Arctic Holocene climate change.
In general, I would like to see a more detailed description of the criteria used to accept or reject studies of Holocene GIC variability in the Arctic, and (somewhat) improved consistency with respect to including/excluding different studies (compare the sections on the Canadian Arctic to the Russian Arctic, for example).

Thank you! We agree that there are many non-lake-based studies that provide complementary information on glacier status that could have been discussed. However, the manuscript is already quite lengthy, and we think that focusing on lake-based glacier reconstructions brings a unique perspective in that they provide continuous records of glacier fluctuations through the Holocene and insights into regional climates that allow for (1) the evaluation of questions such as, when were glaciers smaller than present, and when were summers warmer? And (2) their present status to be placed within a long-term context. This is the first geographically broad synthesis to solely focus on lake-based glacier reconstructions. There have been many additional lake-based records published since the last global review of Holocene glacier fluctuations in 2015 (i.e., Solomina et al., 2015).

We accept all published lake-based Holocene records of glacier fluctuations that clearly define mountain glacier or ice cap status (i.e., specifically if/when they were smaller than present or absent and/or when they regrew in lake catchments) and with sufficient age control to define their status in time. In general, we also make an effort to mention and briefly describe any lake-based Holocene glacier records that we do not include in their respective regional sections so that the additional and valuable information provided is not fully excluded.

We revised 131: We excluded ambiguous records (that do not clearly define when GICs were smaller than present or absent, or when they regrew) and records with poor age control and

included one non–lake–based study from the Russian Arctic (due to the dearth of published glacial lake records there).

Regarding improving the consistency between regions with respect to including/excluding different studies, please see our response to the comment on Page 3, line 87.

Figure 2 is excellent and very clearly explains glacier-lake systems and the stratigraphic records that reflect GIC proximity and how they can differ depending on topography of the lake catchment and size and position of the glacier or ice cap. More reference to these three simplified systems throughout the 'Regional compilations of Holocene GIC records' section would help the reader follow along through this heavy, repetitive section of the manuscript (which must necessarily be repetitive - I do not mean this as a criticism).

Thank you, we agree. We added a symbol to each record in Figures 3-9 (to the right of the site # in panel A) that corresponds to the glacier-lake systems defined in Figure 2.

I do not agree with the title and references in the manuscript that this is a pan-Arctic synthesis, although I do not have an alternative suggestion, unfortunately. The prefix 'pan' means all or involving all members of a group, and there are enormous regions from which no data are included (Canada, Russia), due presumeably to a lack of GICs during the Holocene and/or a lack of studies that fit the criteria of the study (or studies published in english). I will note that an exception was made for Russia due to there being only two lake-based records of GIC fluctuations during the Holocene, but the same exception was not made for the Canadian Arctic, although there were only 5 such studies from a very small corner of the eastern Canadian Arctic Archipelago. There are some, possibly many, studies, which, although might do not fit the criteria perfectly, could possibly have been included to partly fill in this large spatial gap, even just for background context. A quick search and skim resulted in several articles with potential, for example, Holocene fluctuations of Leffert Glacier and nearby outlet glaciers, Ellesmere Island, Nunavut, Canada by W. Blake in Polar Record (2011) and Diatom-based Holocene paleoenvironmental records from continental sites on northeastern Ellesmere Island, high Arctic, Canada by R. Smith in the Journal of Paleolimnology (2002). There are other lake and non-lake-based studies that only cover the late Holocene, for example, but might help to partly fill this spatial gap. If I were to read through these articles more closely, I accept the possibility that they may not fit the criteria, and adding them could also put you in danger of broadening the scope of your study; however, it is somewhat confusing that you made the exception for Russia by including the non-lake based Lubinsky et al. (1999) article. I am not suggesting that you remove this as it does clearly add to this section, but I wonder if there are missed opportunities to fill in the other blank areas on your circumpolar map? With respect to the Arctic Canada section, I will also mention that five lake records from east-central Baffin Island only represents a tiny part of Arctic Canada (the name of the region is thus misleading) - the term used in line 15 to describe this area as the "archipelagos of the eastern Canadian Arctic" is also incorrect.  The title of section 3.2 'Arctic Canada (Baffin Island, northeast Canada)' is much better and more accurate.

We acknowledge that there are large spatial gaps in the lake-based data coverage. However, these gaps do point the community to areas where we are in need of more Holocene lake-based glacier records. To partly address this, we removed the word pan-Arctic from the manuscript and

title. We changed the manuscript title to: Arctic glaciers and ice caps through the Holocene: A circumpolar synthesis of lake–based reconstructions. We think this title better characterizes our study and does not imply that all Arctic regions are covered fully or equally by the available lake-based data. We also acknowledge the uneven coverage by adding the following line to section 2 Data and approach, Line 138: "We note that roughly two–thirds of the available lake–based records are from Greenland and Scandinavia, while other regions (especially geographically large regions, e.g., the Canadian Arctic and Alaska) have less coverage." Finally, we highlighted regions with no lake-based data in Figure 1. We hope this will help point our community to the need for future syntheses of other types of paleoglacier data, and/or future work, in these areas.

Although there are non-lake-based studies that would add to the information presented in our manuscript, we choose to stick with our strict criteria and to only include lake-based glacier studies. We think that adding non-lake-based studies to the Arctic Canada section would too far broaden the scope of our study, as if included for Arctic Canada, we would want to include similar information for other sparsely covered regions (e.g., Alaska) and that vastly expands and alters the scope of our paper. We refer readers instead to existing reviews that incorporate moraine and other evidence, e.g., by Solomina et al. (2015) and Briner et al. (2016).

We revised line 15: Our compilation includes sixty–six lake–based GIC records (plus one non–lake–based record from the Russian Arctic) from seven Arctic regions: Alaska; Baffin Island, northeast Canada; GICs peripheral to the Greenland Ice Sheet; Iceland; the Scandinavian peninsula; Svalbard; and the Russian high Arctic.

The structure of the manuscript is well organized and reads nicely. There are some consistent errors, such as not capitalizing the L in lake when it comes to proper nouns (Igloo Door Lake, not Igloo Door lake, for example) - these and other minor typos, etc., are listed by line below.

Thank you. We capitalized the L in lake in lake names/proper nouns throughout.

Page 1, line 72: homogenously does not work here. Suggest concomitantly instead.

We revised line 72: "Likewise, the onset and rate of summer cooling in the Arctic in the middle–to–late Holocene did not occur concomitantly."

Page 3, line 87: referring to comments above - there is also an apparent dearth of records from the Canadian Arctic (2 lakes for Russia vs. 5 for Canada). Also on this line, see comment above regarding the prefix 'pan'.

We agree that the Canadian Arctic is a geographically large area and is also very poorly covered. We hope that this compilation highlights areas in need of additional lake-based Holocene glacier reconstructions. We only add an additional record to the most poorly covered region, the Russian Arctic. The additional record from the Russian Arctic is especially useful because it combines information from 16 glacier margins in Franz Josef Land, and thus provides the most comprehensive review of Holocene glacier and ice cap fluctuations from the region as a whole.

Page 3, line 89: 'the archipelagos of the eastern Canadian Arctic' is not an appropriate description for east-central Baffin Island.

We replaced 'the archipelagos of the eastern Canadian Arctic' with 'Baffin Island, Canada'

Page 5, line 125: suggest 'to respond' rather than responders

We revised line 125: Second, although relatively quick to respond, it takes some time for GICs to adjust and reach equilibrium or to melt away completely following a shift in climate.

Page 5, line 135: Suggest that 'All available records' should be 'All available lake records accepted according to our criteria' or something similar.

We revised line 135: We also note that in Canada, all lake-based records appropriate for our synthesis are located within the region defined as Arctic Canada South in the RGI.

Page 9, line 261: It might be worth mentioning that both the Penny and Barnes ice caps are remnants of the LIS.

We revised line 261: The island currently hosts the Barnes Ice Cap in central Baffin Island, and Penny Ice Cap, located ~300 km south (both remnants of the Laurentide Ice Sheet), as well as numerous small mountain GICs located along the eastern mountains.

Page 9, pages 264-278 and throughout the manuscript: the 'L' in lake should be capitalized if it is part of a proper noun, unless listed with others, e.g. Yougloo and Igloo Dorr lakes (correct); Igloo Door lake (incorrect). Same for glacier names.

We corrected capitalization in lake and glacier/ice cap names throughout.

Page 10, line 287: fine for consistent language, but I prefer '4 out of 5 of the lake-based records' over '80% of the lake-based records' with such a small number of records here.

We say "at least 80%" because between 10.2-10 ka 2 out of 2 (or 100%) of the records suggest GICs were smaller or absent, while between ~5.9–5.7 ka, 4 out of 5 (or 80%) suggest GICs were smaller or absent.

We revised line 287: At least 80% of the lake–based records from Arctic Canada indicate that GICs were smaller than present or absent between ~10.2–10 ka and ~5.9–5.7 ka, and at least 60% were smaller than present or absent between ~10.2–9.5 ka and between ~8.6–2 ka (though we note that there are few records available from the region, covering a very small geographic area).

Page 11, lines 307-308: '...is highly influenced by various ocean and atmospheric processes, sea ice extent...' is vague and applies to all or almost all of the Arctic regions.

We deleted: 'and is highly influenced by various ocean and atmospheric processes, sea ice extent, and the presence of the GrIS'

Page 12, line 319: 'Persistent glacial input...' is a bit vague. Suggest something more specific. I will also suggest not changing up these terms to describe glacially derived, minerogenic sediment vs. organic-rich sediment too much throughout the manuscript as it is a bit of a distraction.

We revised line 319: Minerogenic sediment input into the lake, implying glacier presence, occurred from ~3.1 ka to present.

Page 12, line 333: Can you include some context regarding the radiocarbon dated reindeer antlers and 'dead plants'?

We revised line 333: Radiocarbon dating of plants and reindeer antlers adjacent to the glacier place bounds on when the site was ice–free or overrun by ice (for study details see Knudsen et al., 2008) and furthermore indicate that the glacier began to expand sometime between ~1.4 and 0.7 ka, when Mittivakkat Glacier advanced towards its maximum LIA extent.

Page 14, line 398: suggest mineral-rich strata rather than mineral-rich units.

We revised line 398: …, except for mineral–rich strata between ~8.8–8 ka and around ~5.7 ka, that may represent brief glacier advances.

Page 20, line 533: Do not understand what you mean by 'several detrital parameters...'

We revised line 533: An exception occurred at ~8.2 ka when minerogenic input abruptly increased, possibly reflecting a reforming glacier.

Page 21, line 575: '...and subsequently has existed continuously...' Awkward description.

We revised line 575: The glacier melted away ~7.3 ka, then reappeared ~6.15 ka and has existed continuously since.

Page 22, line 605: What do you mean by Physical sediment variability?

We revised line 605: A multi-proxy analysis of sediment from a set of glacier–fed lakes show that the ice cap of northern Folgefonna was present between ~11–9.6 ka.

Page 22, line 624: Should be percentage, not percent.

We revised line 624: There are no clear clusters in the timing of GIC regrowth in the middle–to–late Holocene, however, the percentage of GICs smaller or absent starts to decline roughly after ~6 ka, and especially after ~4 ka.

Page 25, line 667: no apostrophe needed in 'lakes'.

Revised line 667: The two lakes which host the highest elevation GICs within their watersheds today …

Page 25, line 669: northernmost is one word

Revised line 669: The northernmost record …

Page 28, line 724: 'The 192...' should be 'the 192...'

Revised line 724: The most prominent archipelagos include: the 192 islands of Franz Josef Land …

Page 28, line 733: It may be that there is a lot known about the Holocene history of GICs in the Russian Arctic, but it has simply not been published in english-language journals?

That is a good point and one that we are not certain about.

We revised line 733: "The Holocene history of GICs in the Russian Arctic is sparsely documented, and we could find only two lake-based records of GIC fluctuations in the English-language literature."

Page 32, line 830: 'other forcings is a bit vague'. Possible to be more specific here?

Kaufman et al., 2004 provides a nice review of these other forcings and feedback mechanisms.

We revised line 830: This marked variability speaks to the complexity of the Arctic climate system's response to insolation, local modulating factors such as ice sheet and ocean influences, and feedback mechanisms (see Kaufman et al., 2004).

Page 34, line 909: I am not familiar with McKay et al (2018) and some other readers might also not be, so I suggest including a little more description of this study to make your point here.

We revised line 909: Similarly, in a review of global Holocene and late Pleistocene alpine glacier fluctuations, Davis et al. (2009) find that glaciers reformed and/or advanced beginning as early as 6.5 ka in some areas. Likewise, using proxy data and climate model simulations, McKay et al. (2018) examine the spatiotemporal patterns, onset, and rate of Neoglacial cooling in the Arctic, and consistent with our inferences, find earliest onset of cooling in Fennoscandia.

Page 35, line 966: Patterns is missing its n.

Revised line 966: Although Arctic–wide patterns emerge, …

**Reviewer #2**

The manuscript presents a compilation of all lake-based reconstructions of local glaciers and ice caps from the circum-Arctic. It describes each of the 65 lake records and synthesize the data in summary figures from each of the seven regions. The data is very skewed towards Greenland and Scandinavia whereas there are less lake records from Russian Arctic and Arctic Canada. Overall, the data is well presented, and it provides an insightful discussion of the results in relation to other climate records. Accordingly, the paper thus falls within the scope of CP.

Thank you. We acknowledge this unevenness of the lake-based glacier records by adding the following sentence to Line 134: We note that roughly two-thirds of the available lake-based records are from Greenland and Scandinavia, while other regions (notably the Russian Arctic, Canadian Arctic, and Alaska) have less coverage.

I only have few suggestions that are meant to improve the manuscript.

Major comments

Figure 2 illustrates very well the type of threshold lakes that have been used in the compilation. I suggest that the authors also include a description of which proxies that are normally used in the studies i.e. XRF core scanning, LOI and magnetic susceptibility etc. It would also be relevant to mention the different types of dating methods and whether the reconstructions rely on macro or bulk 14C dating.

Thank you. We added the following sentence to describe proxies typically used to distinguish glacial and non-glacial sediments. Line 117: Typically, several geochemical and physical properties of sediment are measured (e.g., magnetic susceptibility, major element abundance, grain-size, color, organic matter content, and dry bulk density) and used to distinguish glacial and non-glacial sediments, and to infer glacier activity over time.

We highlight the dating method used in each study/reconstruction (specifically for the oldest Holocene age) by the colored bars in Figures 3-9, i.e., Line 243: Colored bars indicate the oldest reported Holocene age in each record (green=$^{14}$C–dated plant or aquatic macrofossil, yellow=paleomagnetic secular variation (PSV), blue=$^{14}$C dated marine macrofossil, pink=bulk sediment, purple=tephra, black=other).

The data is generally well presented in the summary figures. However, I have some issues with the way some of the lake records have been presented. For example, in figure 3 there are intervals in the lake records that are blank. What does that mean? Is it organic-rich sediments indicating smaller than present, or does it represent a hiatus? Another curious thing is that for many of the records a basal age is below the interpreted section. What is type of sediment is dated and how is it interpreted in relation to glacier history? Leaving intervals blank is not the best solution also because it is not mentioned in the text.

Thank you for requesting clarification. The black sections in figures 3-9 do not necessarily represent a hiatus, but more commonly a time period where the status of the glacier or ice cap is unknown or unclear. For instance, in records that are categorized as 'glacier-lake system 2', organic sediments may indicate a glacier or ice cap that is at present size, smaller, or absent.

Thus, we cannot confidently define the glacier or ice cap status during those periods, and only include when the lake was glacially influenced in our core schematics. In other cases, blank sections indicate times when glacier or ice cap status was unclear or not defined in the original study. To clarify this for readers, we updated the glacier/ice cap status key in panel A for figures 3-9. We also added the following text in the figure 3 caption: Line 246: "… and blank sections denote times when glacier or ice cap status is unknown or unclear."

In cases where the oldest Holocene date is younger than the oldest interpreted sediments, the original study either (1) included a pre-Holocene date, or (2) extended their age model past the oldest date to the base of the core.

Another thing that would help the interpretation is if the individual records are arranged according to which type of threshold lake, they represent i.e. type 1-3.

Although we prefer to arrange the records by glacier/ice cap elevation, we agree with the suggestion to add the "type of threshold lake" and we have added a symbol to each record in Figures 3-9 (to the right of the site # in panel A) that corresponds to the glacier-lake systems (1-3) that we defined in Figure 2. We hope this will help with the reader's interpretation of the individual records.

Minor comments

I would use Early, Middle and Late Holocene as the subdivision of the Holocene has been formally defined by IGS.

We used subdivisions at 8.2 ka (to divide the early and middle Holocene) and 4.2 ka (to divide the middle and late Holocene) following Walker et al. (2018). We added the following sentence in section 2 Data and approach, to explicitly state this: We subdivide the Holocene with the early, middle, and late Holocene beginning at 11.7, 8.2, and 4.2 ka, respectively.

Line 72. Change homogenously to synchronously

We revised line 72: Likewise, the onset and rate of summer cooling in the Arctic in the middle–to–late Holocene did not occur concomitantly.

Line 116. Maybe cite some of the pioneering threshold lake records i.e. Karlen 1981

We revised line 116: The records report local GIC fluctuations through the Holocene reconstructed via analysis of lacustrine sediments from downstream glacial lakes (e.g., Karlén, 1976, 1981; Leonard, 1985, 1986; Karlén and Matthews, 1992).

Line 128. I get the number of lakes to 65 not 66?

In the original paper submission, there are 66 lake-based records plus 1 non-lake-based record from the Russian Arctic. We include a supplementary table listing all the included records.

Line 284-285. Maybe explain why the records are excluded in some more detail.

We accept all published lake-based Holocene records of glacier fluctuations that clearly define mountain glacier or ice cap status (i.e., specifically if/when they were smaller than present or absent and/or when they regrew in lake catchments) and with sufficient age control to define their status in time. In general, we also make an effort to mention and briefly describe any lake-based Holocene glacier records that we do not include in their respective regional sections so that the additional and valuable information provided is not fully excluded.

We revised 131: We excluded ambiguous records (that do not clearly define when GICs were smaller than present or absent, or when they regrew) and records with poor age control and included one non–lake–based study from the Russian Arctic (due to the dearth of published glacial lake records there).

Line 304. I have found a couple of records that have not been included in the compilation. Søndergaard et al 2019 concerning the local ice cap at Qaanaaq and a study concerning Gletscherlukket in SE Greenland by Larsen et al 2021.

Thank you for pointing these records out.

We did not include Søndergaard et al. (2019) in our compilation because the proglacial lake record (referred to as Lake Q3) is used to constrain an integrated signal of ice fluctuations of outlet glaciers of the Greenland Ice Sheet and the Qaanaaq Ice Cap through the Holocene. Since the lake record could not distinguish ice fluctuations of Qaanaaq Ice Cap alone and our study focuses on local glaciers and ice caps distinct from ice sheets (since ice sheets have much longer response times to climate perturbations), we omit this study. However, we have added the following sentences to acknowledge this work. Line 419: Just north of North Ice Cap, Søndergaard et al. (2019) infer the glacial history of outlet glaciers of the Greenland Ice Sheet and the local ice cap, Qaanaaq Ice Cap, via analysis of lake cores from proglacial lake, Lake Q3, geomorphological mapping, $^{10}$Be exposure dating, and $^{14}$C dating of reworked marine mollusks and subfossil plants. The lake record suggests continued glacial meltwater input from its formation at ~7.2 ka until present (Søndergaard et al., 2019). However, since the record cannot distinguish if the sediment deposited in Lake Q3 originates from the Greenland Ice Sheet, the Qaanaaq Ice Cap, or both, and our study focuses only on glaciers and ice caps distinct from the ice sheets, we do not include this record in our compilation.

We did not include Larsen et al. (2021) in our compilation due to insufficient age control. Larsen et al. (2021) present sediment records from two threshold lakes, Lakes XC1423 and XC1424, in southeast Greenland. At present, the lakes do not receive meltwater from the local glacier, Apusiikajik Glacier, as the glacier retreated out of the catchment sometime prior to 1932. During the LIA, the lakes received meltwater when the glacier was at its maximum position. The lake records show that Apusiikajik Glacier receded out of the lake catchments by ~9.6 and remained smaller than its LIA extent until its Late Holocene readvance. However, the timing of the late Holocene readvance is not clear as ages below the silty-clay suggest glacial meltwater input from Apusiikajik Glacier at ~0.5 or 0.2 ka. We have added the following sentences to acknowledge this work. Line 359: Larsen et al. (2021b) present a Holocene record of Apuiikajik Glacier from

two threshold lakes (Lakes XC1423 and XC1424) in southeast Greenland at roughly 63°N. The lakes do not receive glacial meltwater at present as the glacier retreated out of the catchment sometime prior to 1932 CE. The lake records show that Apusiikajik Glacier receded out of the lake catchments by ~9.6 ka and remained smaller than its LIA extent until a readvance in the late Holocene. However, the timing of the readvance is not well constrained, as ages below the silty-clay suggest glacial meltwater input from ~0.5 or 0.2 ka (Larsen et al., 2021b). Thus, we do not include this record in our compilation.

Line 372. I would leave-out the information from the subfossil plants – otherwise similar data from other sites should also be included.

We deleted this point, i.e., we deleted the following line: Radiocarbon dates from subfossil plants provide strong evidence that the ice cap was smaller than present from AD 200 to 1025.

Line 471-478. It is correct that many of the lakes presented in Schomacker et al 2016 are not threshold lakes that receive meltwater at present. However, it is not correct that it does not provide any constraint on the glacier history. One of the lake records receive meltwater from Drangajokull until c. 7.2 ka suggesting the ice cap was larger than present until the Middle Holocene. It would be relevant to add this information.

Thank you for pointing this out. We added the following: Line 503: The lake records also suggest that the northern part of the ice cap was at a similar size or smaller than today by ~10.2 ka whereas the southeast part of the ice cap was larger than today until ~7.8–7.2 ka (Schomacker et al., 2016).

Line 733-742. Why include this information in a compilation of lake records

We included the additional non-lake-based record from the Russian Arctic because of the unique dearth of lake-based information on Holocene glacier fluctuations there. Since Lubinsky et al. (1999) provide the most comprehensive overview of Holocene glacier fluctuations and report information from 16 glacier margins in Franz Josef Land, we decided to include a summary of that evidence in our compilation.

Line 794-796. I am slightly surprised that there are no differences between the lake-based reconstructions and the patterns of GIC fluctuations presented in Solomina et al 2015. It would also be relevant to describe where lake-based records have an advantage and disadvantage compared to other types of proxies used in Solomina et al 2015.

Our compilation shows that the majority of Arctic glaciers and ice caps were smaller than present or absent by ~10 ka, and that most were smaller than present or absent between 7.9-4.5 ka. This finding broadly agrees with Solomina et al., 2015, in that their synthesis finds that in most regions of the mid to high latitudes of the Northern Hemisphere, glaciers were smaller than present or at least equal to their modern sizes between ~8-4 ka. However, the main advantage of our synthesis is that the lake-based records provide unique direct evidence for periods of smaller-than-present ice extent regionally and Arctic-wide.

We revised the final paragraph in section 3.8: Solomina et al. (2015) present the most recent global review of Holocene glacier fluctuations and find that Northern Hemisphere, mid–to–high latitude glaciers were smaller than present or at least equal to their present sizes ~8–4 ka (Solomina et al., 2015). Similarly, a recent study investigating Neoglacial cooling in the Arctic developed a simple index to summarize the relative extent of GICs in Arctic regions (the status of which were derived from the above global review) and reported that GICs were retreated throughout the Arctic ~8–6.5 ka (McKay et al., 2018). The study suggested that this period of uniform retreat, and by inference widespread warmth, can be attributed to high summer insolation combined with the absence of (or lessening influence of) the Laurentide Ice Sheet which lingered well after peak insolation, as late as ~7 ka (McKay et al., 2018). Although our Arctic dataset broadly agrees with other syntheses of Holocene glacier status, our focus on lacustrine archives provides unique direct evidence for periods of smaller–than–present ice extent, and Arctic–wide suggests that most GICs first became smaller than today in the early Holocene.

Line 830. Other forcings – which?

Kaufman et al., 2004 provides a nice review of these other forcings and feedback mechanisms. We revised line 830: This marked variability speaks to the complexity of the Arctic climate system's response to insolation, local modulating factors such as ice sheet and ocean influences, and feedback mechanisms (see Kaufman et al., 2004).

Line 955. Many shrank than today in the Ealy Holocene. I am not sure if that holds. Maybe change to some or even better write x out of y GICs shrank….

Out of the 54 records that provide information on glacier/ice cap status in the early Holocene, 46 (or roughly 85%) showed that the glacier/ice cap first became smaller than present or absent sometime in the early Holocene (i.e., prior to 8.2 ka). For clarity we have changed the text to read "a large majority" instead of "many."